# Activation of DR3 signaling causes loss of ILC3s and exacerbates intestinal inflammation

Jingyu Li [1,7], Wenli Shi [1,7], Hanxiao Sun[2], Yan Ji[1], Yuqin Chen[2], Xiaohuan Guo[3,4,5], Huiming Sheng[2], Jie Shu[2], Liang Zhou[6], Ting Cai[1] & Ju Qiu [1]

TNF-like ligand 1 A (TL1A) and death receptor 3 (DR3) are a ligand-receptor pair involved in the pathogenesis of inflammatory bowel disease. Group 3 innate lymphoid cells (ILC3s) regulate intestinal immunity and highly express DR3. Here, we report that activation of DR3 signaling by an agonistic anti-DR3 antibody increases GM-CSF production from ILC3s through the p38 MAPK pathway. GM-CSF causes accumulation of eosinophils, neutrophils and $CD11b^+CD11c^+$ myeloid cells, resulting in loss of ILC3s from the intestine in an IL-23-dependent manner and exacerbating colitis. Blockade of GM-CSF or IL-23 reverses anti-DR3 antibody-driven ILC3 loss, whereas overexpression of IL-23 induces loss of ILC3s in the absence of GM-CSF. Neutralization of TL1A by soluble DR3 ameliorates both DSS and anti-CD40 antibody-induced colitis. Moreover, ILC3s are required for the deleterious effect of anti-DR3 antibodies on innate colitis. These findings clarify the process and consequences of DR3 signaling-induced intestinal inflammation through regulation of ILC3s.

[1] CAS Key Laboratory of Tissue Microenvironment and Tumor, Shanghai Institute of Nutrition and Health, Shanghai Institutes for Biological Sciences, University of Chinese Academy of Sciences, Chinese Academy of Sciences, Shanghai 200031, China. [2] Tongren Hospital, Shanghai Jiao Tong University School of Medicine, Shanghai 200336, China. [3] Institute for Immunology, Tsinghua University, Beijing 100084, China. [4] Department of Basic Medical Sciences, School of Medicine, Tsinghua University, Beijing 100084, China. [5] Beijing Key Lab for Immunological Research on Chronic Diseases, Tsinghua University, Beijing 100084, China. [6] Department of Infectious Diseases and Immunology, College of Veterinary Medicine, The University of Florida, Gainesville, FL 32608, USA. [7] These authors contributed equally: Jingyu Li, Wenli Shi. Correspondence and requests for materials should be addressed to J.Q. (email: qiuju@sibs.ac.cn)

TNF-like ligand 1A (TL1A) and death receptor 3 (DR3) are a TNF family ligand–receptor pair that is expressed in both mice and humans and has important functions in inflammatory diseases[1]. TL1A is mainly expressed by antigen presenting cells upon stimulation with Toll-like receptor (TLR) ligands or immune complexes[2], whereas DR3 is broadly expressed by lymphocytes, including CD4+ T cells, CD8+ T cells, invariant natural killer T (iNKT) cells, and innate lymphoid cells (ILCs)[1,3]. TL1A/DR3 signaling has been reported to promote the proliferation and cytokine production by subsets of T effector cells and facilitate the progress of inflammatory diseases, such as experimental autoimmune encephalomyelitis (EAE) and allergic pulmonary diseases[1,2,4,5].

In addition to the pro-inflammatory aspect, activation of TL1A/DR3 signaling also has an immunosuppressive role through expansion of T-regulatory cells (Tregs)[4]. This effect can be achieved by administration of an agonistic antibody against DR3 or TL1A-Ig, which is considered to be a therapeutic strategy for autoimmune diseases[6,7]. Activation of DR3 signaling has shown treatment efficacy on multiple inflammatory diseases, such as EAE, lung inflammation, and graft-versus-host diseases (GVHD)[6,8–10], therefore sometimes leading to contradictory conclusions on the effect of TL1A/DR3 with TL1A/DR3-blockade studies. The impact of TL1A/DR3 signaling on inflammation may be determined by environmental factors.

The dual role of TL1A/DR3 signaling in inflammation is also manifested in inflammatory bowel disease (IBD). Mice lacking DR3 or TL1A are susceptible to dextran sulfate sodium (DSS)-induced colitis accompanied by reduced number of Tregs[11]. However, TL1A/DR3 signaling has been predominantly proved by numerous studies to be detrimental in IBD[12–14]. Polymorphisms of Tnfsf15 (encoding TL1A) has been linked to susceptibility to Crohn's disease (CD) by genome-wide association studies in humans[15,16]. Expression of TL1A is enhanced in inflamed tissues of IBD patients[17]. Genetically forced expression of TL1A causes spontaneous small intestinal inflammation featured by overt type 2 innate immune responses[3,18]. Furthermore, neutralization of TL1A in chronic DSS-induced colitis ameliorates the disease, probably by limiting Th1 and Th17 responses[19].

Group 3 innate lymphoid cells (ILC3s) and ILC2s are subsets of ILCs that lack specific T- and B-cell receptors[20]. Abundant numbers of both ILC3s and ILC2s are localized in the intestine of mice, and play crucial roles in intestinal inflammation[21,22]. Notably, DR3 has been found to be highly expressed by all subsets of ILCs[3]. TL1A transgenic mice manifest spontaneous small intestinal inflammation mediated by IL-5 and IL-13 secreted from ILC2s independently of T cells, indicating ILC2s being key targets for TL1A-driven inflammation[3,18]. In the intestine, TL1A is reported to be mainly derived from $CX_3CR1^+$ mononuclear phagocytes[23]. TL1A has been shown to promote GM-CSF and IL-22 expression from ILC3s in synergy with IL-1β and IL-23, which contributes to the immune defensive function of $CX_3CR1^+$ mononuclear phagocytes in intestinal infection[23]. Whether ILC3s participate in TL1A-mediated intestinal inflammation is unclear.

Under the steady-state of the intestine, ILC3s have been found to be a primary source for GM-CSF that is involved in intestinal inflammation through various mechanisms[24]. GM-CSF supports the induction of intestinal Tregs through accumulation of TGF-β-producing CD11b+CD103+ dendritic cells[24]. On the other hand, GM-CSF can be highly pathogenic in colitis by activating eosinophils and enhancing cytokine production from eosinophils[25]. Another report has shown that blockade of GM-CSF ameliorates α-CD40-induced innate colitis and orchestrates mobilization of ILC3s during colitis[26]. Notably, reduced number of ILC3s in α-CD40-induced innate colitis in mouse models is compatible with the observation that fraction of ILC3s is reduced in inflamed intestine of CD patients compared with non-inflamed controls[26–28]. Although the conversion of ILC3s to ILC1s has been demonstrated to be a reason for this phenomenon, other molecular mechanisms underlying the dynamics of ILC3s in IBDs may exist.

In this study, we report that α-DR3 exacerbates colitis through stimulation of ILC3s, while finally induces loss of ILC3s in the large intestine. Agonistic antibody targeting DR3 (α-DR3) robustly drives loss of ILC3s, which is accompanied by infiltration of pro-inflammatory myeloid cells. Mechanistically, α-DR3 promotes GM-CSF production from ILC3s, which leads to the accumulation of eosinophils, neutrophils, and IL-23-producing CD11b+CD11c+ cells. Blockade of either GM-CSF or IL-23 reverses α-DR3-driven ILC3 loss, whereas IL-23 eliminates ILC3s from the large intestine in the absence of GM-CSF. α-DR3 exacerbates DSS-induced colitis independently of the adaptive immune system. Strikingly, the exacerbation of DSS-induced innate colitis by α-DR3 is not observed in the absence of ILC3s, suggesting a critical role of ILC3s in initiating the inflammatory storm. Transcriptome analysis reveals upregulation of MAPK cascade-related genes in ILC3s upon α-DR3 treatment. We have further proved that p38 signaling is essential for GM-CSF production by ILC3s induced by TL1A/DR3 signaling in both mice and humans. Our finding uncovers the molecular mechanism underlying the intestinal pathology mediated through TL1A/DR3 signaling in ILC3s.

## Results

**α-DR3 induces reduction of ILC3 numbers in the large intestine.** DR3 has been found to be highly expressed by subsets of ILCs in the intestine[3]. To examine the effect of DR3 signaling on intestinal ILCs, we treated mice with an α-DR3 agonistic antibody (4C12), which has been shown to boost Tregs and may be implicated in treatment of autoimmune diseases. Tregs were efficiently expanded in the large intestine by two injections of 2.5 μg of α-DR3 every other day (Supplementary Fig. 1A, B). In addition, we observed a significantly decreased proportion of NKp46−ILC3s among the Lineage− (Lin−) cells in LPLs (lamina propria lymphocytes) of the large intestine (Fig. 1a, b; Supplementary Fig. 1C)[29]. There was also a mild trend toward a reduction in percentage of NCR+ILC3s among Lin− cells (Fig. 1a, c), while percentages of ILC2s and ILC1s in Lin− cells were not affected by α-DR3 treatment (Fig. 1a, d). The total number of ILC3s was also reduced (Fig. 1e). The mRNA expression of IL-22 in large intestinal LPLs was decreased upon α-DR3 treatment probably due to loss of ILC3s, which are the major sources of IL-22 (Fig. 1f)[30,31].

To identify if α-DR3-driven loss of ILC3s was dependent on the adaptive immune system, we administered α-DR3 to Rag1−/− mice which lack T and B cells. The expression of DR3 was similar in subsets of ILCs from Rag1−/− mice compared with wild-type mice (Supplementary Fig. 1D). Strikingly, the total number of ILC3s, as well as proportions of NCR+ILC3s and NKp46−ILC3s among Lin− cells, was dramatically decreased upon α-DR3 treatment (Fig. 1g–i; Supplementary Fig. 1E). Consistently, absolute number of IL-22-producing ILC3s was decreased in α-DR3-treated mice (Fig. 1j). Similar effect on ILC3s was achieved by overexpression of TL1A in Rag1−/− mice using hydrodynamic injection (Fig. 1k–n). Using Rorc^gfp/+Rag1−/− mice with a GFP reporter to indicate the expression of RORγt[32], we analyzed the distribution of ILC3s (GFP+ cells) in the large intestine by immunofluorescence staining of GFP. In α-DR3-treated mice, we observed a reduction in the number of cryptopatches where ILC3s are typically localized (Fig. 1o, p)[33]. Although the total number of ILC3s per analyzed cryptopatch was not affected by α-

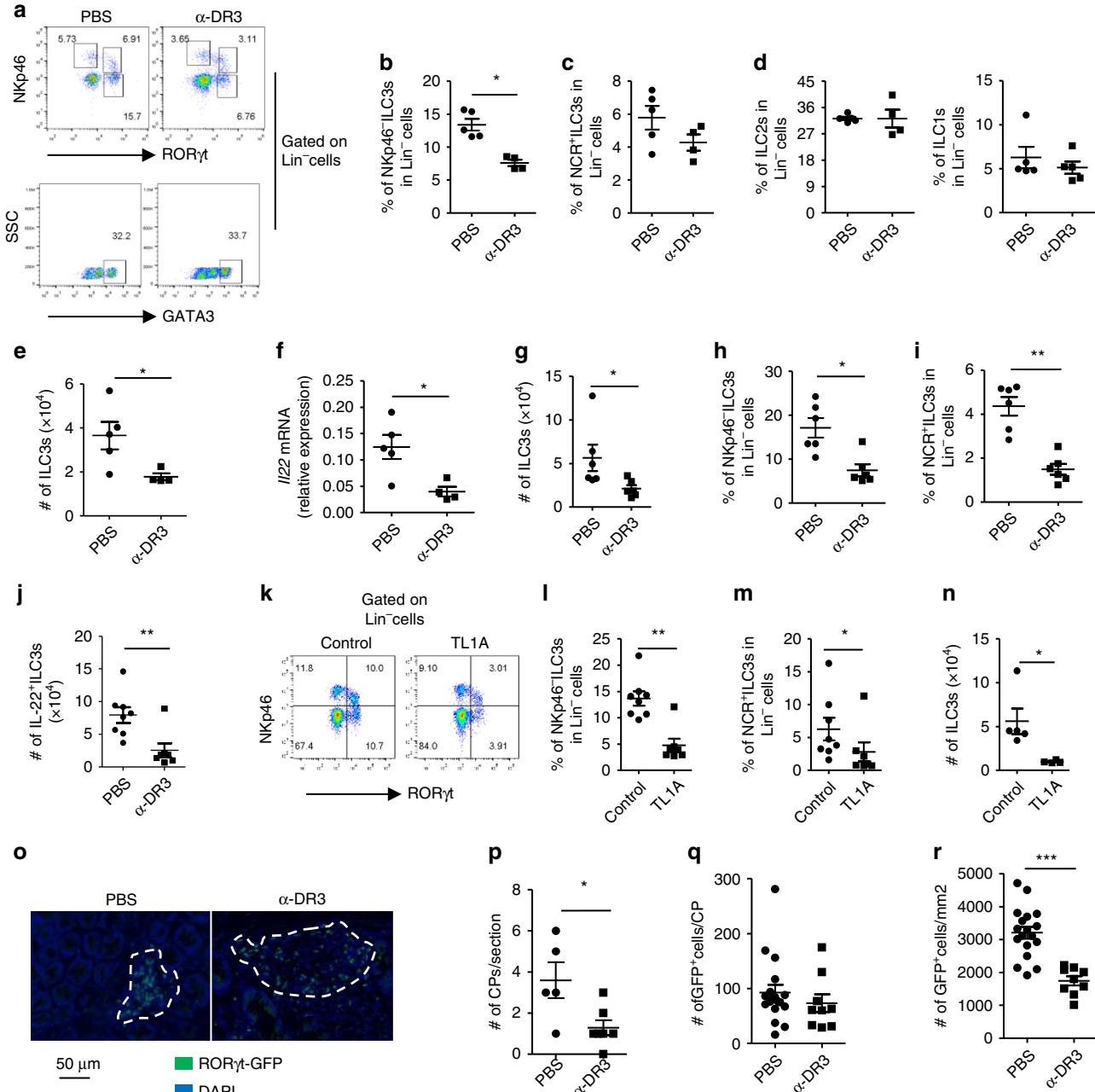

**Fig. 1** α-DR3 induces reduction of ILC3s in the large intestine independent of the adaptive immune system. Wild-type (**a–f**), *Rag1*−/− mice (**g–j**), or *Rag1*−/− *Rorc*gfp/+ mice (**o–r**) were treated with α-DR3 antibody (4C12), and large intestinal lamina propria lymphocytes (LPLs) were isolated for analysis 4 days later. **k–n** *Rag1*−/− mice were treated with 10 μg of the TL1A or control plasmid DNA through hydrodynamic injection, and large intestinal LPLs were isolated for analysis 4 days later. **a–e**, **g–n** Expression of RORγt, NKp46, GATA-3, IL-22, and Lineage markers (Lin, CD3, B220, CD11b, and CD11c) was analyzed by flow cytometry. Percentages of NKp46−ILC3 (Lin−NKp46−RORγt+) (**b**, **h**, **l**), NCR+ILC3 (Lin−NKp46+RORγt+) (**c**, **m**), ILC2 (Lin−GATA3high) (**d**), and ILC1 (Lin−NKp46+RORγt−) (**d**) gated on Lin− cells are shown. **e**, **g**, **n** The total numbers of ILC3s in indicated groups are shown. **j** For detection of IL-22, large intestinal LPLs were treated with brefeldin A 2 h before cells were harvested for analysis by flow cytometry. Absolute numbers of IL-22+ILC3s (Lin−RORγt+IL-22+ cells) are shown. **f** mRNA expression of IL-22 in large intestinal LPLs was analyzed by real-time RT-PCR. **o–r** Expression of RORγt-GFP and DAPI in colon sections was analyzed by immunofluorescence. **o** Representative structures of cryptopatches are shown. **p** Numbers of crytopatches (CP) are shown. **q** Numbers of RORγt-GFP+ cells (ILC3s) per CPs were counted and shown. **r** Density of RORγt-GFP+ cells (ILC3s) per CPs was calculated and shown. The data are means ± SEM. **a–r** The data are representative of at least three independent experiments. Source data are provided as a Source Data File

DR3 treatment (Fig. 1o, q), the structure of cryptopatches was looser, leading to reduced density of ILC3s in individual cryptopatches (Fig. 1o, r). The above data suggest α-DR3-induced loss of ILC3s is independent of the adaptive immune system. Control IgG had no observed effect on loss of ILC3s compared with PBS (Supplementary Fig. 1F–K), we thus utilized PBS as a control in the following experiments.

**α-DR3-induced ILC3 loss is not a result of cell apoptosis in situ.** The percentage of Ki67+ ILC3s was enhanced upon treatment with α-DR3 (Fig. 2a, b), suggesting there is no defect in proliferation of ILC3s upon α-DR3 treatment[34]. Activation of DR3 has been reported to induce cell apoptosis[35]. However, level of cleaved caspase 3, an indicator of cell apoptosis[36], was not elevated in ILC3s at early and late time point of α-DR3-treated mice (Supplementary

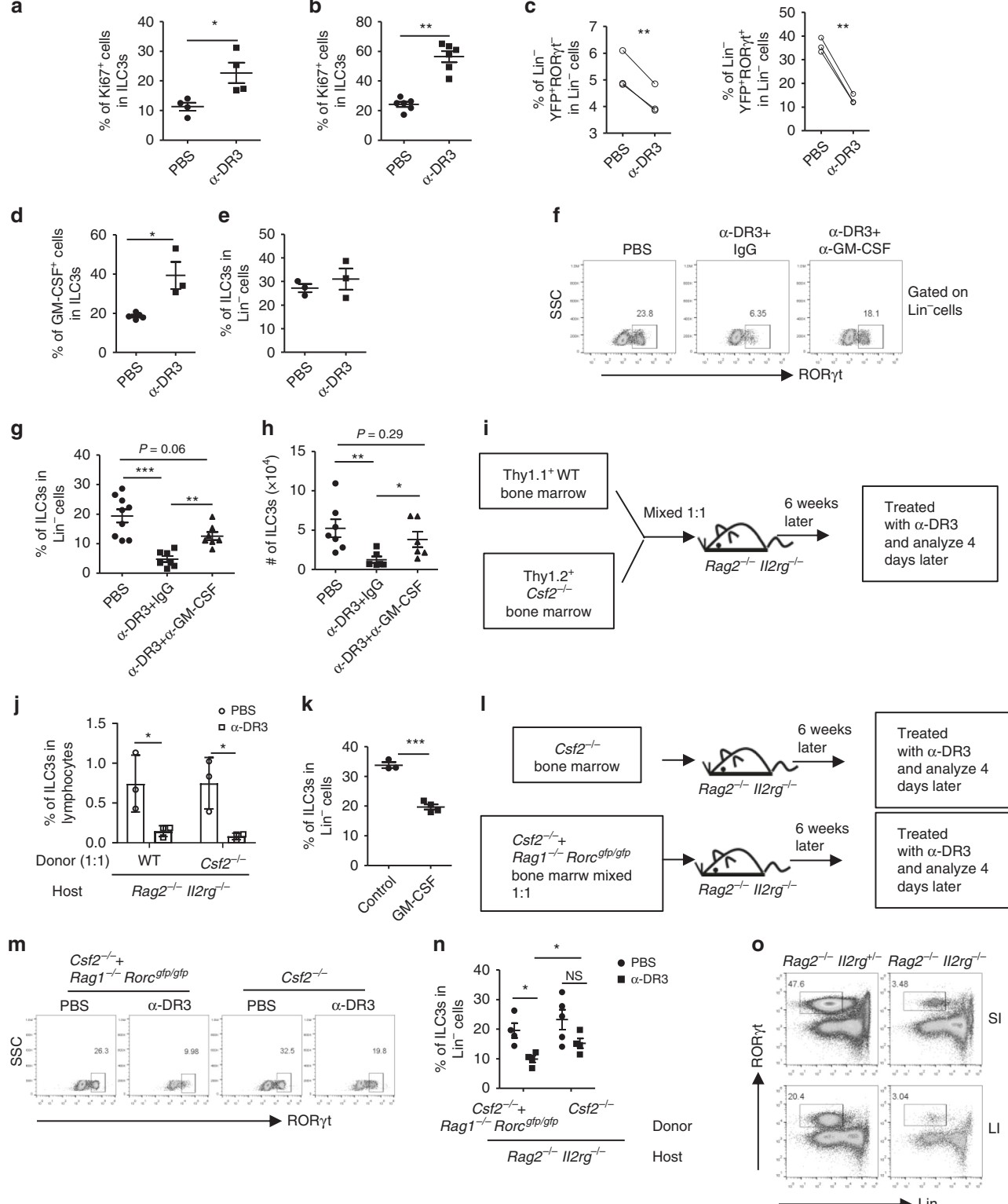

Fig. 2A). Furthermore, treatment of mice with zVAD-FMK, a pan-caspase inhibitor, failed to block the reduction of ILC3s induced by α-DR3 (Supplementary Fig. 2B, C). The above data indicate that α-DR3-driven loss of ILC3s is not due to cell apoptosis.

The conversion of ILC3s to ILC1s can occur under the inflammatory status, and has been suggested to contribute to the proportional change of ILC1 and NKp44+ILC3 in CD patients[37,38]. To determine if α-DR3-induced loss of ILC3s was due to a fate conversion of ILC3s, we injected α-DR3 to *Rag1*−/−

*Rorc-creRosa26^{stop-YFP}* mice, which could be used to track the fate of ILC3s by examining the expression of YFP (representing both current and historic ILC3s) and RORγt (representing current ILC3s) (Fig. 2c; Supplementary Fig. 2D)[39]. A reduction of both Lin−YFP+RORγt− cells (exILC3s) and Lin−YFP+RORγt+ cells (current ILC3s) was observed in α-DR3-treated mice (Fig. 2c; Supplementary Fig. 2D). The level of exILC3s in the α-DR3-treated mice did not compensate for the loss of current ILC3s (Fig. 2c; Supplementary Fig. 2D). Consistently, no compensatory

**Fig. 2** Autocrine and paracrine GM-CSF induced by α-DR3 is critical for ILC3 loss. $Rorc^{gfp/+}$ (**a**), $Rag1^{-/-}Rorc^{gfp/+}$ (**b**), or $Rag1^{-/-}Rorc\text{-}creRosa26^{stop\text{-}YFP}$ mice (**c**) were treated with α-DR3, and large intestinal LPLs were analyzed 4 days later. **a–c** Expression of Ki67, Lin, RORγt, and YFP was analyzed. **a**, **b** Percentages of Ki67+ cells in ILC3s are shown. **c** Percentages of Lin−YFP+RORγt− and Lin−YFP+RORγt+ cells in Lin− cells are shown. **d**, **e** $Rag1^{-/-}$ mice were treated with α-DR3 antibody, and large intestinal LPLs were isolated for analysis 24 h later. **f–h** Mice were treated with α-DR3 with (α-DR3+α-GM-CSF) or without (α-DR3+IgG) injection of neutralization antibody for GM-CSF. **i**, **j**, **l–n** $Rag2^{-/-}Il2rg^{-/-}$ mice were half-lethally irradiated and transferred with Thy1.1+ wild-type bone marrow (BM) and Thy1.2+$Csf2^{-/-}$ BM mixed at 1:1 ratio (**i**, **j**), $Csf2^{-/-}$ BM, or $Csf2^{-/-}$ BM and $Rag1^{-/-}Rorc^{gfp/gfp}$ BM mixed at 1:1 ratio (**l–n**). **i**, **l** Protocols for bone marrow transfer. **k** $Rag1^{-/-}$ mice were treated with 10 μg of GM-CSF or control plasmid DNA through hydrodynamic injection, and large intestinal LPLs were isolated for analysis 4 days later. **d–o** Expression of Lin, RORγt, and GM-CSF in large intestinal LPLs was analyzed. **d** Percentages of GM-CSF gated on ILC3s (Lin−RORγt+) are shown. **f**, **m** Expression of RORγt gated on Lin− cells were analyzed. **e**, **g**, **k**, **n** Percentages of ILC3s (Lin−RORγt+) gated on Lin− cells are shown. **j** Percentages of ILC3s from wild-type origin (Lin−Thy1.1+RORγt+) and $Csf2^{-/-}$ origin (Lin−Thy1.1− RORγt+) in total lymphocytes are shown. **h** The total number of ILC3s are shown. The data are means ± SEM. **o** Small and large intestinal LPLs from 6-week-old littermate $Rag2^{-/-}Il2rg^{+/-}$ and $Rag2^{-/-}Il2rg^{-/-}$ mice were isolated, and expression of lineage markers and RORγt was analyzed. Percentages of ILC3s (Lin−RORγt+) gated on live lymphocytes are shown. **c**, **n** Statistical analyses were performed with paired $t$ test. **a–o** The data are representative of at least two independent experiments. Source data are provided as a Source Data File

---

increase in numbers of ILC1s, ILC2s, or Lin−non-ILCs was found upon α-DR3 treatment, although there was a trend toward a proportionally increased level of ILC2s among Lin− cells, probably due to primary loss of ILC3s (Supplementary Fig. 2E, F). The above data imply that fate conversion is not a key mechanism for the reduction of ILC3s induced by α-DR3 at least in situ.

**GM-CSF induced by α-DR3 is critical for ILC3 loss.** Mobilization of ILC3s induced by GM-CSF has been found to cause loss of ILC3s in innate colitis[26]. A previous study has shown that TL1A promoted GM-CSF production by ILC3s in vitro[23]. When $Rag1^{-/-}$ mice were treated with α-DR3, GM-CSF production by ILC3s was significantly enhanced 24 h after α-DR3 treatment, before loss of ILC3s was observed (Fig. 2d, e). Strikingly, blockade of GM-CSF with a neutralizing antibody efficiently prevented loss of ILC3s that was triggered by α-DR3 (Fig. 2f–h). The above data suggest that GM-CSF induced by α-DR3 is critical for loss of ILC3s in the intestine.

GM-CSF could originate from various types of immune cells during inflammation[24,40,41]. We thus questioned if paracrine source of GM-CSF could drive loss of ILC3s induced by α-DR3. We constructed bone marrow chimeric mouse on ILC-deficient $Rag2^{-/-}Il2rg^{-/-}$ recipients by mixing donor bone marrows from Thy1.1+wild-type and Thy1.2+$Csf2^{-/-}$ mice at a 1:1 ratio (Fig. 2i). In the above system, environmental GM-CSF could be supplemented by wild-type bone marrows. After treatment with α-DR3, a significant reduction of ILC3s from $Csf2^{-/-}$ donors was observed at a level similar to ILC3s from the wild-type donors, suggesting paracrine GM-CSF induced by α-DR3 is sufficient to induce loss of $Csf2^{-/-}$ILC3s (Fig. 2j). In align with this scenario, systemic overexpression of GM-CSF by hydrodynamic injection also resulted in loss of ILC3s in the large intestine (Fig. 2k).

In the $Rag2^{-/-}Il2rg^{-/-}$ recipients that were reconstituted with $Csf2^{-/-}$ bone marrow cells, no significant loss of ILC3s was induced by α-DR3, although a trend to a decrease was observed, probably due to few GM-CSF-sufficient ILC3s developmentally independent of the common gamma chain in the intestine (Fig. 2m–o). The above phenomenon confirms that hematopoietic cell-derived GM-CSF triggered by α-DR3 signaling is required for ILC3 loss. Supplementation of $Csf2^{-/-}$ bone marrow cells with $Rag1^{-/-}Rorc^{gfp/gfp}$ bone marrow cells resulted in dramatic loss of ILC3s from $Csf2^{-/-}$ donors in response to α-DR3 (Fig. 2m, n). This suggests that innate non-ILC3 sources of GM-CSF from the hematopoietic compartment triggered by α-DR3 could induce reduction of ILC3s in the intestine. In summary, α-DR3-induced GM-CSF from both ILC3s and non-ILC3s collectively drive the loss of ILC3s.

**IL-23 is important for α-DR3-induced loss of ILC3s.** Receptor for GM-CSF is typically expressed by myeloid cells but not ILC3s, so elimination of ILC3s by GM-CSF may be indirect. Indeed, we found that the mRNA expression of both $Csf2ra$ and $Csfr2b$ was much lower in ILC3s than in the CD11b+ myeloid cells (Fig. 3a, b), and $Csf2rb$ was not detected on ILC3s (Fig. 3a, b). A previous study suggests that IL-23, which could be produced by macrophage or dendritic cells, is important for pathology of α-CD40-induced colitis, in which ILC3 loss occurs[26]. We found that the mRNA expression of $Il23a$ and $Il12b$, genes encoding two subunits of IL-23, was enhanced upon α-DR3 treatment (Fig. 3c). When we neutralized p40, a shared subunit by IL-23 and IL-12, loss of ILC3s driven by α-DR3 was significantly blocked (Fig. 3d–f). Consistently, blockade of IL-23 with α-p19, but not blockade of IL-12 with α-p75 neutralizing antibodies, reversed loss of ILC3s induced by α-DR3, as indicated by percentages and absolute numbers of ILC3s (Fig. 3g–j). To further investigate the effect of IL-23 or IL-12 on loss of ILC3s, we overexpressed IL-23 or IL-12 in vivo through hydrodynamic injection. Consistently, forced expression of IL-23 but not IL-12 led to a dramatic reduction of ILC3s in the large intestine (Fig. 3k, l), suggesting a key role for IL-23 in α-DR3-driven ILC3 loss.

The above findings showed that both GM-CSF and p40 were required for α-DR3-induced ILC3 loss (Figs. 2g, h, 3e, f), and that overexpression of either GM-CSF or IL-23 alone could cause reduction of ILC3s (Figs. 2k, 3k, l). However, GM-CSF overexpression failed to induce ILC3 loss when p40 was blocked (Fig. 3m). Conversely, IL-23 overexpression caused comparably profound loss of ILC3s in $Csf2^{+/-}$ and $Csf2^{-/-}$ mice (Fig. 3n, o). This suggests that overexpression of IL-23 could induce loss of ILC3s in a GM-CSF-independent mechanism.

**Upregulation of IL-23 by α-DR3 is dependent on GM-CSF.** Upon α-DR3 treatment, loss of ILC3s was found to be accompanied by infiltration of CD11b+ myeloid cells (Fig. 4a–d). In wild-type mice, the total numbers and percentages of CD11b+ CD11c+ cells and eosinophils (CD11b+CD11c−Siglec-F+) in live cells were dramatically enhanced (Fig. 4a, b; Supplementary Fig. 3). In $Rag1^{-/-}$ mice, the numbers of CD11b+CD11c+cells, eosinophils, and neutrophils (CD11b+CD11c−Ly6G+) were all increased upon α-DR3 treatment (Fig. 4c; Supplementary Fig. 3). Strikingly, blockade of GM-CSF suppressed the infiltration of CD11b+CD11c+cells, eosinophils, and neutrophils in $Rag1^{-/-}$ mice as indicated by the decreased proportion of these cells among live cells (Fig. 4d; Supplementary Fig. 3). In addition, induction of $Il23a$ and $Il12b$ mRNA expression in large intestinal LPLs was significantly suppressed when GM-CSF was neutralized (Fig. 4e). We further purified different subsets of cells that have been reported to be sources of IL-23 in the intestine

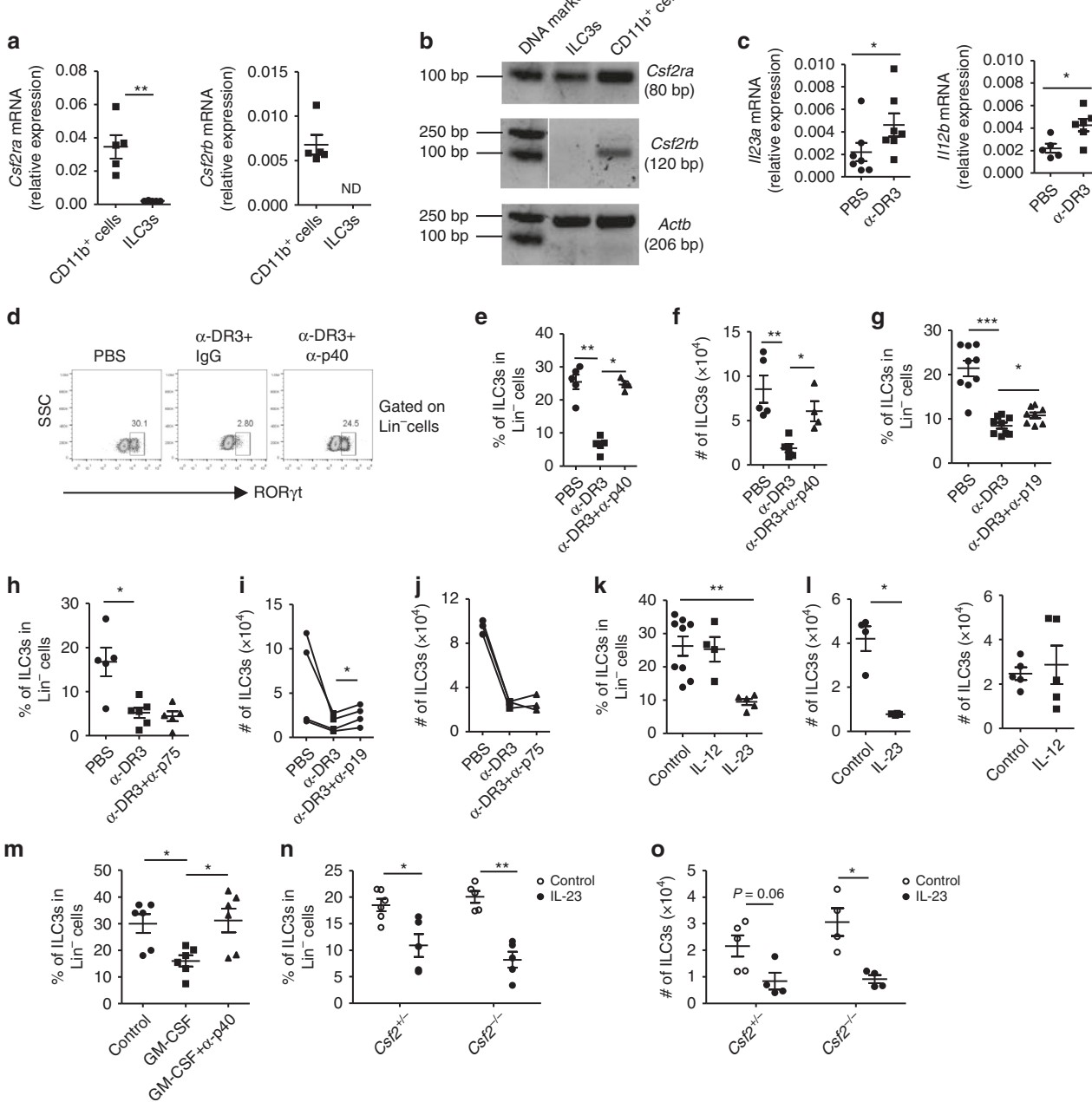

**Fig. 3** IL-23 is important for α-DR3-induced loss of ILC3s. **a** The mRNA expression of *Csf2ra* and *Csf2rb* in purified CD11b+ cells and ILC3s (Lin⁻GFP+) from large intestinal LPLs of *Rag1⁻/⁻Rorc^gfp/+* mice was analyzed by real-time RT-PCR. **b** The cDNA product from real-time RT-PCR analysis in (**a**) was analyzed by electrophoresis. **c** *Rag1⁻/⁻* mice were treated with α-DR3 antibody. The mRNA expression of *Il23a* and *Il12b* in large intestinal LPLs was analyzed by real-time RT-PCR. **d–j** *Rag1⁻/⁻* mice were treated with α-DR3 in the presence or absence (control IgG) of neutralizing antibody for p40, p19, or p75. Large intestinal LPLs were isolated for analysis on day 4 post α-DR3 treatment. **k, l** *Rag1⁻/⁻* mice were treated with 10 μg of IL-12, IL-23, or control plasmid DNA through hydrodynamic injection, and large intestinal LPLs were isolated for analysis 4 days later. **m** *Rag1⁻/⁻* mice were treated with 10 μg of GM-CSF or control plasmid DNA through hydrodynamic injection with (GM-CSF+α-p40) or without (GM-CSF+IgG) i.p. injection of neutralization antibody for p40 simultaneously. **n, o** Littermate mice with indicated genotypes were treated with 10 μg of IL-23 or control plasmid DNA through hydrodynamic injection. **k–o** Large intestinal LPLs were isolated for analysis 4 days later. **d–o** Expression of Lin and RORγt were analyzed by flow cytometry. **d** Expression of RORγt gated on Lin⁻ cells is shown. **e, g, h, k, m, n** Percentages of ILC3s (Lin⁻RORγt+) gated on Lin⁻ cells are shown. **f, i, j, l, o** The total numbers of ILC3s are shown. **a, c, e–h, k–o** The data are means ± SEM. **i, j** Dots indicate average number of ILC3s per mouse from every batch of the experiment. The data from the same batch of the experiment are connected with a solid line. Statistical analyses were performed with paired *t* test. **a–o** The data are representative of at least two independent experiments. Source data are provided as a Source Data File

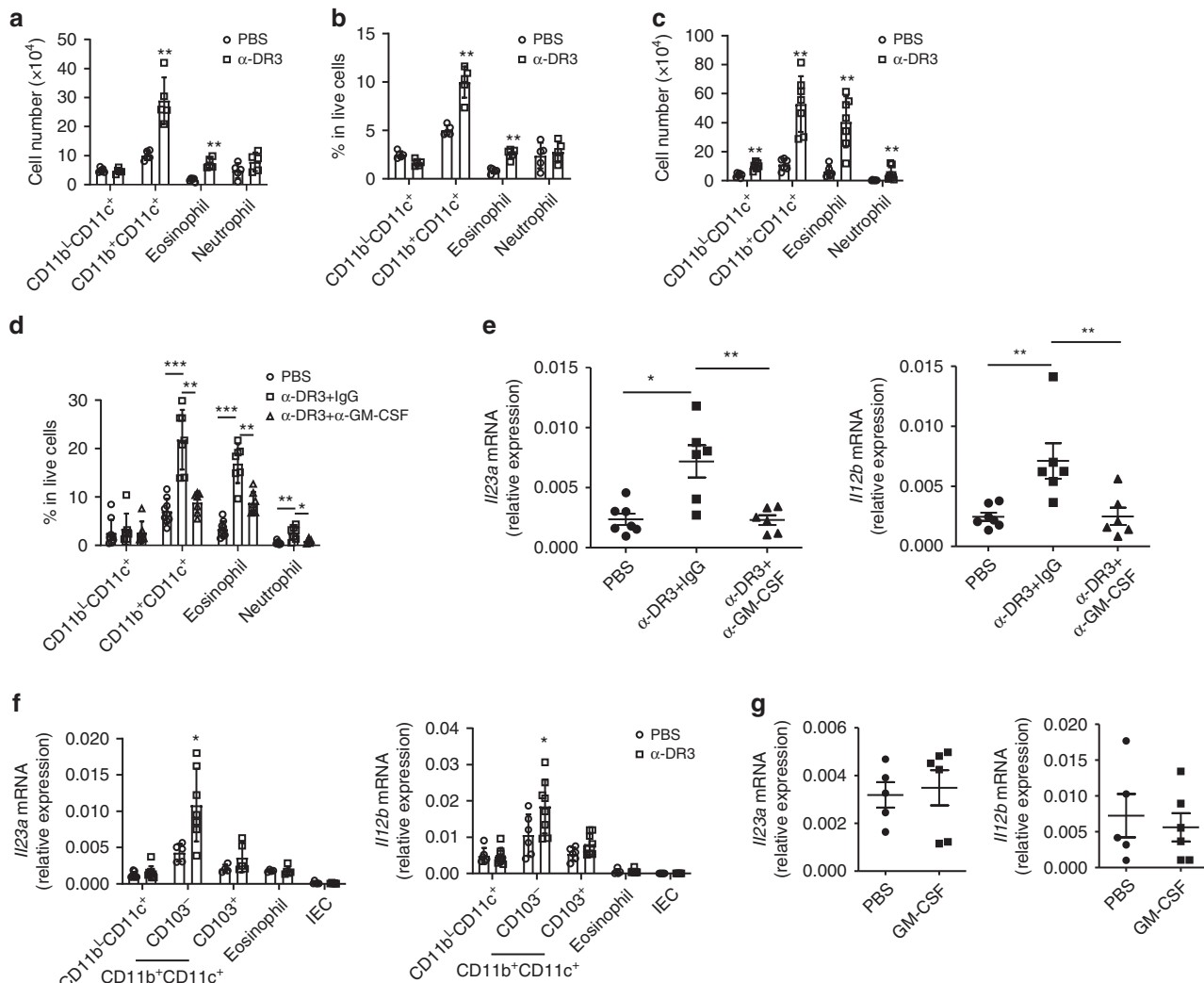

**Fig. 4** GM-CSF mediates enhancement of IL-23 induced by α-DR3. Wild-type (**a**, **b**) or *Rag1⁻/⁻* mice (**c–f**) were treated with α-DR3. **d**, **e** *Rag1⁻/⁻* mice were treated with α-DR3 with (α-DR3+α-GM-CSF) or without (α-DR3+IgG) injection of neutralization antibody for GM-CSF. **a–f** Large intestinal LPLs were isolated for analysis 4 days after α-DR3 treatment. **a–d** Expression of CD11b, CD11c, CD103, Siglec-F, and Ly6G were analyzed by flow cytometry. **a**, **c** The total numbers of CD11b^L CD11c^+ (CD11b^L CD11c^+CD103^+) cells, CD11b^+CD11c^+ cells, eosinophils (CD11b^+CD11c^−Siglec-F^+), and neutrophils (CD11b^+ CD11c^−Ly6G^+) gated on live cells are shown. **b**, **d** Percentages of indicated populations gated on live cells defined in (**a**, **c**) are shown. **e** The mRNA expression of *Il23a* and *Il12b* in large intestinal LPLs was analyzed by real-time RT-PCR. **f** The mRNA expression of *Il23a* and *Il12b* in large intestinal epithelial cells (IEC), purified CD11b^L CD11c^+ (CD11b^L CD11c^+CD103^+), CD11b^+CD11c^+CD103^+ cells, CD11b^+CD11c^+CD103^− cells, and eosinophils (CD11b^+CD11c^−Siglec-F^+) from large intestinal LPLs was analyzed by real-time RT-PCR. **g** CD11b^+CD11c^+CD103^− cells were purified from large intestinal LPLs of *Rag1⁻/⁻* mice under the steady state and treated with PBS or GM-CSF (20 ng/ml) for 4 h. The mRNA expression of *Il23a* and *Il12b* was analyzed by real-time RT-PCR. **f** Statistical analyses were performed with Student's *t* test. **a–g** The data are means ± SEM. The data are representative of at least two independent experiments. Source data are provided as a Source Data File

(Supplementary Fig. 4A)[23,24,42–44]. Real-time RT-PCR analysis revealed that CD11b^+CD11c^+CD103^− cells, which include previously described sources of IL-1β/IL-23/TL1A[23,24,42], were predominantly enriched for both *Il23a* and *Il12b* mRNA expression (Fig. 4f; Supplementary Fig. 4A). Notably, CD11b^+CD11c^+ CD103^− cells but no other analyzed types of cells showed increased *Il23a* and *Il12b* mRNA expression in α-DR3-treated group than control (Fig. 4f; Supplementary Fig. 4A). Although a previous study suggested a direct role for GM-CSF in enhancing the mRNA expression of IL-23[45], GM-CSF had no observed effect on *Il23a* and *Il12b* mRNA expression in purified CD11b^+CD11c^+ CD103^− cells in vitro (Fig. 4g; Supplementary Fig. 4A). Together, these data indicate that α-DR3 leads to enhanced IL-23 expression in the intestine by accumulation of CD11b^+IL-23-producing myeloid cells through GM-CSF. A previous study has shown that

IL-23 promotes GM-CSF production from ILC3s[26]. Since both IL-23 and GM-CSF were upregulated by α-DR3 treatment (Figs. 2d, 3c), it is likely that IL-23 and GM-CSF collaboratively drive the loss of ILC3s in an autocrine loop during α-DR3-induced inflammation.

**α-DR3 stimulates GM-CSF production from ILC3s through p38.** To determine the effect of α-DR3 on ILC3s at a genome-wide scale, we purified ILC3s from α-DR3 or PBS-treated *Rorc^gfp/+Rag1⁻/⁻* mice and performed transcriptome sequencing. Total of 515 and 609 genes were found to be upregulated and downregulated, respectively, for more than 1.5-fold in ILC3s from α-DR3-treated mice compared with the control group (Supplementary Data 1 and Supplementary Fig. 4B). Gene ontology enrichment analysis of upregulated targets in the α-DR3 treatment

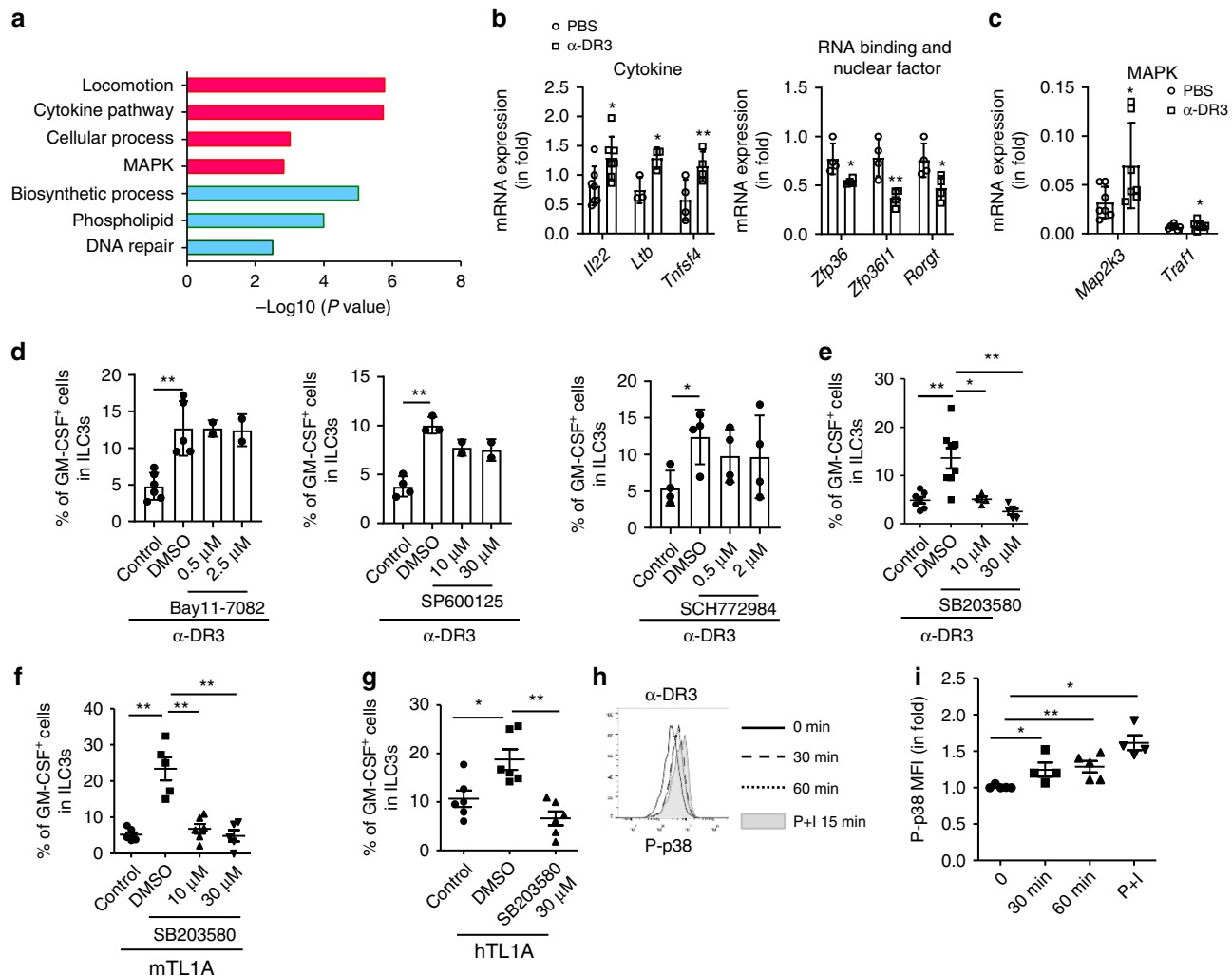

**Fig. 5** α-DR3 stimulates GM-CSF production from ILC3s through p38 signaling. **a, b** $Rag1^{-/-}Rorc^{gfp/+}$ mice were treated with 1 μg of α-DR3 once, and large intestinal LPLs were isolated 3 days later. **a** Duplicates mRNA of FACS purified ILC3s (Lin⁻GFP⁺) cells were exacted and subjected to genome-wide analysis (RNA-seq). Gene ontology enrichment analyses were performed on upregulated genes (red) and downregulated genes (green) of more than 1.5-fold in the α-DR3-treated group, and representative enriched biological processes are shown. **b** Relative expression of indicated genes in purified ILC3 (Lin⁻GFP⁺) was analyzed by real-time RT-PCR and normalized to the PBS group. **c** ILC3s (Thy1.2^high CD45^intermediate) cells were purified from $Rag1^{-/-}$ mice and treated with PBS or α-DR3 (250 ng/ml) for 18 h. Expression of indicated genes was analyzed by real-time RT-PCR. **d–f** Large intestinal LPLs were isolated from $Rag1^{-/-}$ mice. Cells were treated with PBS, TL1A (100 ng/ml), and α-DR3 (250 ng/ml) in the presence or absence (DMSO) of inhibitors for NF-kB (Bay11-7082), JNK (SP600125), ERK (SCH772984), and p38 (SB203580) for 18 h. Brefeldin A was added for the last 2 h before analysis. Expression of GM-CSF in ILC3s (Lin⁻RORγt⁺) was analyzed by flow cytometry and are shown. **g** Human tonsil lymphocytes were treated with PBS or human recombinant TL1A (100 ng/ml) in the presence or absence (DMSO) of inhibitors for p38 (SB203580) for 18 h. Brefeldin A was added for the last 2 h before analysis. Percentages of GM-CSF expression in ILC3s (Lin⁻CD127⁺CD117⁺) was analyzed and are shown. **h, i** Large intestinal LPLs from $Rag1^{-/-}$ mice were treated with α-DR3 or PMA plus ionomycin for indicated time interval. **h** Expression of phosphorylated p38 (p-p38) gated on ILC3s (Lin⁻RORγt⁺) was analyzed and are shown in histogram. **i** MFI (mean fluorescence intensity) of p-p38 gated on ILC3s was normalized to the control group (time point 0), and fold change is shown. Statistical analyses were performed with paired $t$ test (**b**) and Student's $t$ test (**c**). **e–g, i** The data are means ± SEM. **b–i** The data are representative of at least two independent experiments. Source data are provided as a Source Data File

group revealed overrepresentation of biological processes, including cytokine mediated signaling pathway, cellular defense response, locomotion, and MAPK cascade (Fig. 5a; Supplementary Figs. 4B, 5A). The downregulated genes were enriched in phospholipid metabolic process, DNA repair, and biosynthetic processes (Fig. 5a; Supplementary Fig. 4B). We also manually categorized a series of downregulated genes functionally related to immune regulation (Supplementary Figs. 4B, 5B). Notably, confirmation of the downregulated expression of *RORgt* and upregulated expression of *Tnfsf4* in ILC3s from α-DR3-treated mice by real-time RT-PCR was consistent with a previously reported role for TL1A on splenic Lti cells in vitro (Fig. 5b)[46]. Cytokines

upregulated in ILC3s of α-DR3-treated mice, including *Ltb* and *Il22*, indicated enhanced function of ILC3s to recruit neutrophils (Fig. 5b)[47,48]. A group of upregulated genes belonging to gene category of locomotion, consisting of chemokine receptors, Rho GTPase, and growth factors, suggested modulation of cell migration activity by DR3 signaling (Supplementary Figs. 4B, 5A, 5C)[49]. The upregulation of *Ccr7* together with the downregulation of *Ccr9* and *Itgb7* reflected a suppression of the retention of α-DR3-treated ILC3s in the gut (Supplementary Fig. 5C)[50,51]. However, the differential expression of CCR7 protein in ILC3s in α-DR3-treated mice was inconsistent with the change at mRNA level (Supplementary Fig. 5D). We observed no accumulation of

ILC3s in other organs, including the small intestine, mesenteric lymph nodes, spleen, peritoneal cavity, or blood, suggesting ILC3s did not preferentially or specifically translocate to one of the above places (Supplementary Fig. 5E, F). Interestingly, upregulated genes in the cellular defense response group contained a cluster of members encoding MHCII (Supplementary Figs. 4B, 5A)[52,53]. Flow-cytometric analysis confirmed the enhanced expression of MHCII at the protein level in ILC3s of α-DR3-treated mice (Supplementary Fig. 5G, H). We also found that the mRNA expression of Zfp36 and Zfp36l1, tristetraprolin molecules playing regulatory roles in inflammation, was decreased in ILC3s of α-DR3-treated mice (Fig. 5b)[54].

In accordance with upregulation of Ccr6 mRNA expression in ILC3s upon α-DR3 treatment, we observed a trend toward a proportional increase of CCR6+NKp46− ILC3s and a trend toward a decrease of NCR+ILC3s and CCR6−NKp46− ILC3s among the total ILC3s (Supplementary Fig. 6A, B). But the absolute numbers of all subsets of ILC3s were reduced (Supplementary Fig. 6C). Notably, the upregulation of MHCII expression was mainly on NCR+ILC3s and CCR6−NKp46− ILC3s (Supplementary Fig. 6D), whereas the upregulation of IL-22 expression was mainly on NCR+ILC3s and CCR6+NKp46− ILC3s (Supplementary Fig. 6E). Enhancement of OX40L expression was found on all subsets of ILC3s upon treatment of α-DR3 (Supplementary Fig. 6F).

The upregulation of several MAPK cascade genes implied activation of MAPK signaling, which was consistent with previously reported downstream signaling of DR3 through MAPK (Supplementary Figs. 4B, 5A)[55]. We have confirmed by real-time RT-PCR that the expression Map2k3 and Traf1 was significantly upregulated when purified ILC3s were treated with α-DR3 in vitro (Fig. 5c; Supplementary Fig. 4C). We then investigated if GM-CSF expression, which was a key event for inflammatory storm driven by α-DR3, was regulated by the MAPK pathway. Using a series of small molecule inhibitors, we found that inhibitors of p38 (SB203580), but not NF-κB (Bay11-7082), JNK (SP600125), or ERK (SCH772984), dramatically suppressed α-DR3-induced GM-CSF expression in ILC3s induced by α-DR3 (Fig. 5d, e). TL1A-induced GM-CSF expression by ILC3s was similarly inhibited by SB203580 (Fig. 5f). Importantly, TL1A-induced GM-CSF in human tonsil ILC3s was also suppressed by SB203580, suggesting a conserved regulation of GM-CSF by p38 in mice and humans (Fig. 5g). Consistent with this observation, an increased level of phosphorylated p38 was observed in ILC3s within 1 h of α-DR3 treatment (Fig. 5h, i). SB203580 had no cytotoxic effects on human or mouse ILC3s, as indicated by the comparable percentages of live lymphocytes, similar number of ILC3s and no reduction of proportions of ILC3s among lymphocytes in SB203580-treated groups compared with control groups (Supplementary Fig. 7). Therefore, the suppression of GM-CSF expression in ILC3s by SB203580 was not due to cytotoxicity. The above data suggest that α-DR3 signals through the p38 pathway to induce the expression of GM-CSF by ILC3s.

**DR3-Fc treatment ameliorates α-CD40-induced colitis**. During α-CD40-induced colitis, the mRNA expression of TL1A was significantly elevated (Fig. 6a). To evaluate the role of TL1A/DR3 signaling in α-CD40-induced intestinal pathology, we neutralized TL1A by treating mice with DR3-Fc, a synthesized decoy receptor for TL1A by expression of DR3 conjugated with mouse-Fc (Fig. 6b–j). We found that the DR3-Fc-treated mice lost less body weight and recovered sooner than vehicle or IgG-treated group in α-CD40-induced colitis (Fig. 6b). Moreover, colon length in DR3-Fc-treated group was longer (Fig. 6c, d).

Percentages of neutrophils in the LPLs were decreased in DR3-Fc-treated mice (Fig. 6e). Histological analysis using hematoxylin and eosin (H&E) staining showed less severe inflammation in DR3-Fc-treated mice (Fig. 6f, g). Consistent with the function of α-DR3 in promoting GM-CSF expression by ILC3s, we observed reduced GM-CSF expression by ILC3s in DR3-Fc-treated mice 24 h after α-CD40 injection (Fig. 6h). The above data indicate a pathogenic role for TL1A in α-CD40-induced colitis.

Loss of ILC3s has been reported in α-CD40-induced colitis in previous study[26]. We found that blockade of TL1A by DR3-Fc significantly restored the cell numbers and percentages of ILC3s in lymphocytes compared with control (Fig. 6i, j), whereas no effect of DR3-Fc on percentages and numbers of ILC3s was observed under the steady state (Supplementary Fig. 8A, B). These results suggest an essential role for TL1A/DR3 signaling in ILC3 loss during innate colitis.

**α-DR3 exacerbates innate colitis in an ILC3-dependent manner**. A previous study showed that blockade of TL1A ameliorated chronic DSS-induced colitis by suppression of Th1 and Th17 responses in wild-type mice[19]. In DSS-induced colitis in Rag1−/− mice, we found that DR3-Fc-treated mice lost less weight and had increased colon length (Fig. 7a, b; Supplementary Fig. 8C). The total numbers of neutrophils were less in DR3-Fc-treated mice (Fig. 7c). This suggests that TL1A/DR3 signaling is pathogenic in DSS-induced colitis in the absence of the adaptive immune system. Consistent with the protective effect of DR3-Fc in colitis, α-DR3-exacerbated DSS-induced colitis in wild-type mice manifested by more rapid rate of weight loss, despite a promotive effect on immune-suppressive Tregs (Fig. 7d; Supplementary Fig. 8D). In DSS-induced colitis, α-DR3-treated mice showed shorter colon lengths (Fig. 7e; Supplementary Fig. 8E), enhanced numbers of neutrophils and eosinophils (Fig. 7f), and exacerbated intestinal pathology characterized by damage to the epithelial cells and infiltration of leukocytes than control group (Fig. 7g; Supplementary Fig. 8F). In Rag1−/− mice, α-DR3 treatment similarly caused a more rapid rate of weight loss (Fig. 7i), shorter colons (Fig. 7j; Supplementary Fig. 8G), increased number of neutrophils and eosinophils in lamina propria (Fig. 7k), as well as more severe intestinal pathology during DSS-induced colitis compared with controls (Fig. 7l; Supplementary Fig. 8H). Furthermore, in both wild-type and Rag1−/− mice, the exacerbation of colitis in α-DR3-treated group was accompanied by loss of ILC3s (Fig. 7h, m). Therefore, α-DR3-exacerbated DSS-induced colitis independently of the adaptive immune system.

Previous studies have shown that enhanced TL1A/DR3 signaling results in small intestinal inflammation by promoting the function of ILC2s[3,18]. To delineate the role of ILC3s in TL1A/DR3-induced inflammation, we administered α-DR3 and induced DSS colitis in littermate Rag1−/−Rorc^gfp/+ and Rag1−/−Rorc^gfp/gfp mice. The latter lacked ILC3s. Without α-DR3 treatment, Rag1−/−Rorc^gfp/+ and Rag1−/−Rorc^gfp/gfp mice exhibited similar percentages of weight loss (Fig. 7n), colon length (Fig. 7o; Supplementary Fig. 8I), and intestinal pathology as indicated by the infiltration of neutrophils/eosinophils and histological changes in the colon tissues (Fig. 7p, q; Supplementary Fig. 8J). However, α-DR3 exacerbated the colitis indicated by above indexes in Rag1−/−Rorc^gfp/+, but not in Rag1−/−Rorc^gfp/gfp mice, suggesting ILC3s are required for the pathogenic effect of α-DR3 in DSS-induced innate colitis (Fig. 7n–q; Supplementary Fig. 8I, J). Loss of ILC3s was observed in Rag1−/−Rorc^gfp/+ mice upon α-DR3 treatment (Supplementary Fig. 8K). However, the activation of ILC3s by α-DR3 signaling at an early phase may serve as a prerequisite for exacerbation of colitis by α-DR3. We therefore

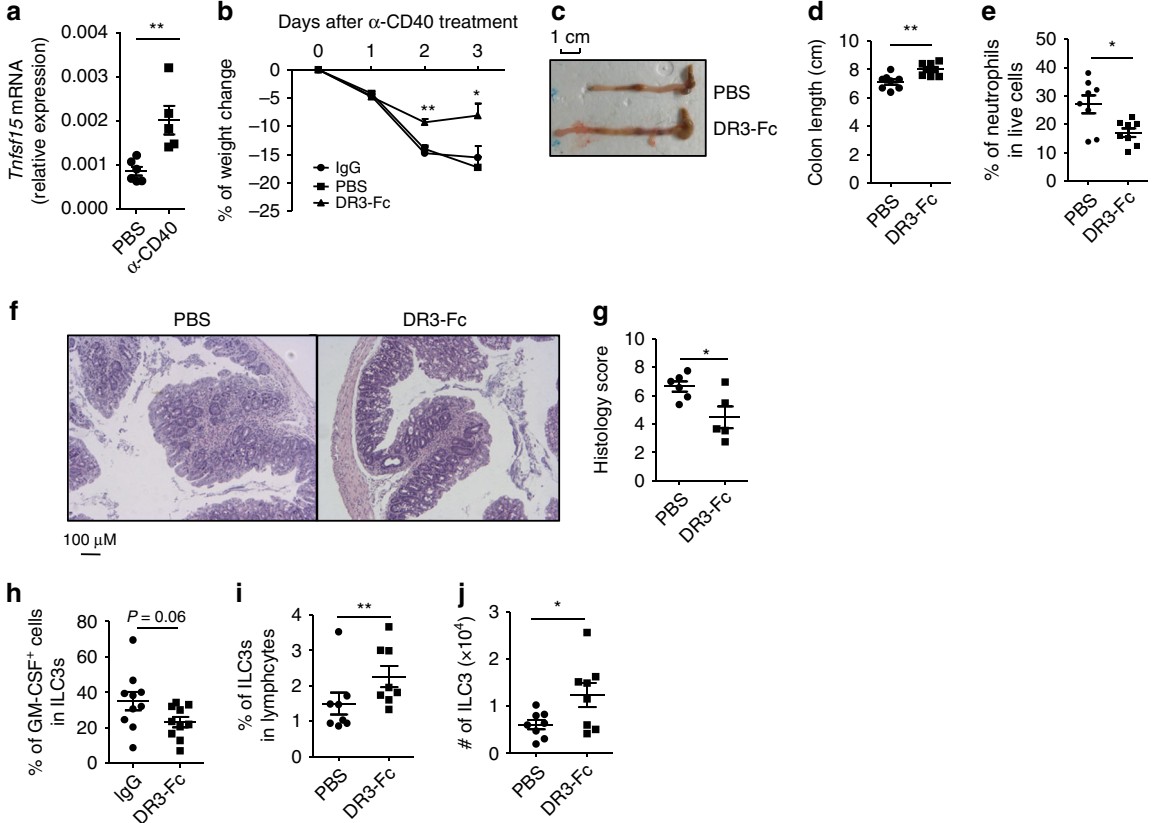

**Fig. 6** DR3-Fc treatment ameliorates α-CD40-induced colitis. Female littermate $Rag1^{-/-}$ mice were injected with 50 μg of α-CD40. **a** Expression of $Tnfsf15$ mRNA in large intestinal LPLs was analyzed by real-time RT-PCR 48 h after treatment. **b–j** 200 μg of DR3-Fc or control IgG or PBS were i.p. administered daily from day 0. **c–g**, **i**, **j** Mice were killed for analysis on day 3. **b** Percentages of weight change are shown. Representative picture of large intestine (**c**) and lengths of colons (**d**) from indicated groups are shown. **e**, **h**, **i** Large intestinal LPLs were isolated, and the expression of Lin, RORγt, CD11b, and Ly6G was analyzed by flow cytometry. **e** Percentages of neutrophils (CD11b$^+$Ly6G$^+$) in live cells are shown. **i** Percentages of ILC3s (Lin$^-$RORγt$^+$) gated on lymphocytes are shown. **j** The total numbers of ILC3s were calculated and are shown. **h** Mice were killed for analysis 24 h later after α-CD40 injection, and large intestinal LPLs were isolated. Expression of Lin, RORγt, and GM-CSF was analyzed by flow cytmetry. Percentages of GM-CSF$^+$ cells in ILC3s are shown. Statistical analysis was performed with Student's t test. **f** Paraffin-embedded colon sections were stained with hematoxylin and eosin (H&E). Scale bar is 100 μm. **g** Histological scores were evaluated and shown. The data are means ±;SEM. **a–j** The data are representative of at least two independent experiments. Source data are provided as a Source Data File

conclude ILC3s are indispensable for α-DR3-exacerbated DSS-induced colitis in $Rag1^{-/-}$ mice.

## Discussion

DR3 exhibits broad expression profile among lymphocytes and regulates inflammation in dual directions[1,3]. The activation of DR3 signaling by an agonistic antibody has been proved to be effective in alleviating GVHD and lung allergy in mouse models by promoting Tregs[6,8–10]. The activation of DR3 signaling has also been indicated to be applied in immuno-cancer therapy by facilitating the CD8$^+$T-cell responses[56].

However, TL1A/DR3 signaling is considered to play a pathogenic role in IBD through various mechanisms, which brings forward a high risk for the exacerbation of intestinal inflammation by activation of DR3 signaling[12]. In this study, we have found that activation of DR3 signaling leads to exacerbation of the colitis through targeting ILC3s, which were finally eliminated from the intestine through GM-CSF/IL-23-dependent mechanism. In wild-type mice, targets for α-DR3 are broad and include subsets of CD4$^+$ T effector cells, CD8$^+$ T cells, and ILCs[1,3], and the exacerbating effect of α-DR3 may be a reflection of multifaceted immune responses influenced by DR3 signaling. In $Rag1^{-/-}$ mice, targets for DR3 are more limited, and this model provided us a simpler system to delineate the impact of α-DR3 on ILC3s.

A previous study showed that TL1A or DR3-deficient mice were susceptible to DSS-induced colitis due to impaired maintenance of Tregs[11]. In addition, a recent study reported that mice with genetic deletion of DR3 specifically on ILC3s were more susceptible to DSS-induced colitis, mainly due to decreased IL-22 from ILC3s[57]. Our data, together with a previous report, suggest that TL1A/DR3 signaling plays a pathogenic role in DSS-induced colitis[19]. The discrepancies among the above findings on the role of DR3 in innate colitis may be due to differential experimental settings using genetic-deficient mice compared with transient blockade of DR3 signaling using TL1A neutralization antibody or DR3-Fc. Genetic deficiency of TL1A or DR3 could be considered as a long-term prevention protocol for colitis, which could not be completely represented with antibody blockade. It is possible that in mice with genetic deficiency of DR3 on ILC3s, constantly suboptimal level of IL-22 from birth makes the individual more susceptible to exacerbated colitis led by lack of IL-22. Whereas in antibody treatment experiments, blockade of TL1A or DR3 started after onset of DSS-induced colitis, before which no possibly impaired epithelial integrity due to lack of IL-22 occurred. Therefore, blockade of TL1A/DR3 using antibodies resulted in ameliorated colitis likely by suppressing the pathogenicity of ILC3s. In the future, it will be necessary to test whether a prevention protocol by long-term blockade of TL1A/DR3 using antibodies would increase the susceptibility to intestinal inflammation.

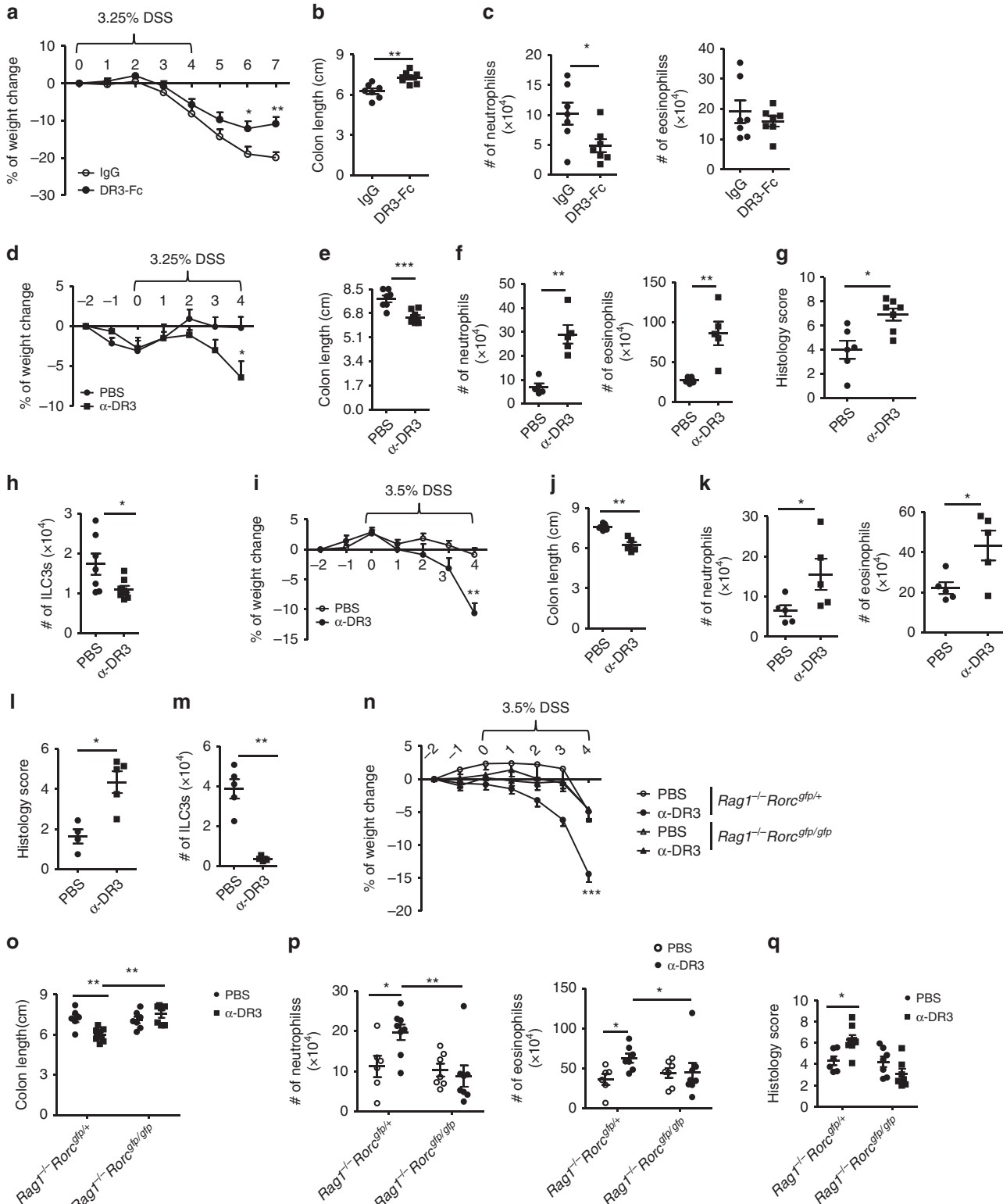

**Fig. 7** α-DR3 exacerbates DSS-induced innate colitis through an ILC3-dependent manner. Littermate $Rag1^{-/-}$ (**a**–**c**, **i**–**m**), wild-type (**d**–**h**), $Rag1^{-/-}Rorc^{gfp/+}$, and $Rag1^{-/-}Rorc^{gfp/gfp}$ mice (**n**–**q**) were fed with DSS in drinking water as indicated in **a**, **d**, **i**, and **n**. **a**–**c** 200 μg of DR3-Fc or control IgG were i.p. administered daily from day 3–6. **d**–**q** 2.5 μg (for wild-type mice) or 1 μg (for mice of $Rag1^{-/-}$ background) α-DR3 antibody or PBS was i.p. injected to mice on day −2, 0, and 2 post DSS treatment. **a**, **d**, **i**, **n** Percentages of weight change are shown. **b**, **e**, **j**, **o** Lengths of colons are shown. **c**, **f**, **h**, **k**, **m**, **p** Large intestinal LPLs were isolated on day 4 (except for on day 7 for c) post DSS treatment, and the expression of CD11b, Ly6G, Siglec-F, Lin, and RORγt were analyzed by flow cytometry. **c**, **f**, **k**, **p** Absolute numbers of neutrophils (CD11b+Ly6G+) and eosinophils (CD11b+SiglecF+) were analyzed by flow cytometry, calculated, and shown. **h**, **m** Absolute numbers of ILC3s (Lin−RORγt+) were analyzed by flow cytometry, calculated, and shown. **g**, **l**, **q** Paraffin-embedded colon sections were subjected to H&E staining, and histological scores are evaluated and shown. The data are means ± SEM. **a**–**q** The data are representative of at least two independent experiments. Source data are provided as a Source Data File

We have demonstrated that α-DR3 treatment results in the accumulation of IL-23-producing myeloid cells through GM-CSF and promotes the function of CD11b⁺CD11c⁺CD103⁻ cells to express IL-23 at the mRNA level. A previous study has shown that GM-CSF production from ILC3s can be enhanced by IL-23[26]. It is possible that DR3 signaling works in synergy with IL-23 to induce GM-CSF production from ILC3s, generating a positive feedback loop for the progression of inflammation. Furthermore, CD11c⁺MHCII^high myeloid cells have been found to localize closely with ILC3s in cryptopatches or lymphoid follicles during colitis, which may allow for prompt crosstalk between the two[26,33]. Except for GM-CSF, IL-23 could also enhance the production of IL-17, IL-22, and IFN-γ from ILC3s, which further facilitate the recruitment of pro-inflammatory myeloid cells[58]. Therefore, although GM-CSF could be from other sources, on the basis that they are IL-23R-expressing cells and neighbors of IL-23-producing cells, ILC3s appeared to be an irreplaceable effector population in α-DR3-exacerbated innate colitis.

We have shown that IL-23 induces loss of ILC3s from the intestine even in the absence of GM-CSF. Consistent with this observation, another study found that IL-23 administration caused loss of ILC3s from the small intestine[59]. The reduction of proportion of IL-22-producing ILC3s is phenotypically in accordance with the situation in CD patients[27,28]. IL-23-driven loss of ILC3s may be a critical though not unique mechanism for this phenomenon. The correlations among TL1A, GM-CSF, IL-23 expression, and reduction of IL-22-producing ILC3s in patients with IBD needs to be further investigated. Interestingly, a recent study demonstrated that IL-23 mediates the expulsion of T, B, and NK cells in the mouse model of lung tumor[60]. Therefore, IL-23 may directly or indirectly orchestrate profiles of chemotaxis in local environment and result in reorganization of the composition of immune cells. Further understanding the molecular mechanism underlying exclusion of immune cells by IL-23 will provide insights into pathogenesis of IBD and tumors.

We demonstrated that loss of ILC3s induced by α-DR3 is less likely due to apoptosis or fate conversion in situ. It is possible that ILC3s undergo other forms of cell death independently of caspase 3 such as necroptosis[61]. Mice with genetic deficiency for key signaling molecules controlling other forms of cell death will help to test this possibility. Search for ILC3s in multiple tissues failed to locate accumulation of ILC3s in the small intestine, mesenteric lymph nodes, spleen, peritoneal cavity, or blood. Although fate conversion of ILC3s was not found in situ, it is also possible that ILC3s first migrate to other organs and then fate conversion of ILC3s takes place. A thorough spatial–temporal inspection for ILC3s locally and in various organs, especially with the facilitation of *Kaede* mouse and intravital microscopy, is required for further elucidation of the mechanisms for loss of ILC3s[62,63].

A previous study suggests that mobilization of ILC3s out of the cryptopatches may play a role in coordination and perpetuation of the inflammation in the gut[26]. However, final reduction of ILC3s can result in contraction of the inflammation, which works as an efficient feedback mechanism to control overt inflammatory responses. The strategies we used for blocking loss of ILC3s using α-GM-CSF, α-p19, α-p40, or DR3-Fc in this study may also simultaneously ameliorate colitis by limiting the pathogenicity of ILC3s or other mechanisms[26,64]. This makes it difficult for us to assess the consequence of loss of ILC3s per se, without affecting cytokine production of ILC3s. One would suspect abnormal retention of pro-inflammatory ILC3s would lead to persistent inflammation and severe tissue damage, which would be important to investigate in the future.

## Methods

**Mice.** Wild-type mice were purchased from Shanghai SLAC Laboratory Animal Co. Rag1⁻/⁻, Csf2⁻/⁻, Rorc^gfp/gfp, Rorc-cre, Rosa26^stop-YFP, and Thy1.1 mice were purchased from Jackson laboratory. Rag2⁻/⁻Il2rg⁻/⁻ mice were purchased from Taconic Biosciences. Mice used for in vivo studies were littermate controlled and were 6–10 weeks old. Both male and female mice were used unless otherwise noted. All mice used in this study are on C57BL/6 background, and were maintained in specific pathogen-free conditions. In DSS-induced colitis, α-CD40-induced colitis and other in vivo experiments, littermate mice were co-housed since weaning and randomly separated into control and treatment groups to avoid variation in distribution of microflora. All mice were fed with a plain commercial diet (Silaikang, Shanghai). Mice were housed in corn-cob-bedding cages in a room with the light–dark cycle (lights on at 6:00 and off at 18:00). All animal experiments were performed in compliance with the guide for the care and use of laboratory animals, and were approved by the institutional biomedical research ethics committee of the Shanghai Institutes for Biological Sciences, Chinese Academy of Sciences.

**In vivo treatment of mice with antibodies.** Mice on a wild-type or Rag1⁻/⁻ background were i.p. injected with agonistic antibody against DR3 (4C12, Biolegend) at 2.5 μg and 1 μg, respectively, every 2 days. Control mice were injected with same amount of Isotype Hamster IgG (Biolegend) or PBS. Unless otherwise noted, mice were killed for analysis 4 days after the first injection. For some experiments, rat IgG (Sangon Biotech) or neutralization antibodies targeting GM-CSF (MP1-22E9, Bioxcel), p40 (C17.8, Bioxcel), p19 (G23-8, Thermo Fisher Scientific), and p75 (R2-9A5, Bioxcel) were injected at a dose of 250 μg on day 0 and day 2. Mice were killed for analysis on day 4 upon α-DR3 treatment. Same protocol of blocking antibody treatment was performed on day 0 and day 2 of hydrodynamic injection experiments.

**Hydrodynamic gene delivery.** Control vector (pRK) and pRK-IL-23 have been previously described. pRK-GM-CSF and pRK-TL1A were generated by cloning the coding sequence of mouse GM-CSF and TL1A to pRK vector, respectively. Generation of pRK-IL-12 was performed according to previously published method[65]: coding sequence of Il12b (the stop codon was deleted) and Il12a (the first 22 amino acids were deleted) was linked by a (Gly₄Ser₃)₃ linker and cloned to the pRK vector. Plasmid DNA was introduced into mice using a hydrodynamic tail vein injection. Briefly, 10 μg of DNA/mouse was diluted in 1.5 to 2.0 ml of TransIT-EE Hydrodynamic Delivery Solution (Mirus) at 0.1 ml/g body weight. The DNA solution was injected into mice through the tail vein using a 27-gauge needle within a time period of 5 to 10 s. Hydrodynamic injection was performed only once, and mice were killed 4 days after injection.

**Bone marrow transfer.** Rag2⁻/⁻Il2rg⁻/⁻ mice were irradiated at 550 rads once, and immediately transferred with $5 \times 10^6$ donor bone marrow cells. Chimeric mice were used for further experiments 6 weeks after transfer.

**Isolation of intestinal LPLs and intestinal epithelial cells.** Small or large intestines were dissected, fat tissues and peyer's patches were removed. Intestines were cut open longitudinally and washed in PBS. Intestines were then cut into 3-cm-long pieces, washed, and shaken in PBS containing 1 mM DTT for 10 min at RT. Intestines were incubated with shaking in PBS containing 30 mM EDTA and 10 mM HEPES at 37 °C for 10 min for two cycles. Supernatant from the first round of EDTA was saved as intestinal epithelial cells. The tissues were then digested in the RPMI1640 medium (Thermo Fisher Scientific) containing DNase I (150 μg/ml, Sigma) and collagenase VIII (200 U/ml, Sigma) at 37 °C in a 5% $CO_2$ incubator for 1.5 h. The digested tissues were homogenized by vigorous shaking and passed through 100-μm cell strainer. Mononuclear cells were then harvested from the interphase of an 80 and 40% Percoll gradient after a spin at 2500 rpm for 20 min at RT.

**Cell suspension preparation from different tissues.** Peritoneal lavage cells were isolated by flushing the peritoneal cavity with 10 ml of PBS. Cell suspensions were prepared from the spleen, and mesenteric lymph nodes by gentle mechanical disruption and passed through a 50-μm nylon mesh. Blood was collected by the heart punctures. Mononuclear cells from the blood were isolated from interface of Ficoll-Paque density-gradient centrifugation at 2500 rpm for 20 min at RT.

**Flow cytometry.** Anti-mouse CD16/32 antibody was used to block the nonspecific binding to Fc receptors before all surface stainings. Dead cells were stained with live and dead violet viability kit (Invitrogen), and were gated out in analysis. Antibodies used for regular flow cytometry are listed in Supplementary Table 1. α-p-p38 was from Cell signaling Technology. For nuclear stainings, cells were fixed and permeabilized using a Mouse Regulatory T Cell Staining Kit (Thermo Fisher Scientific). BD Cytofix/Cytoperm™ kit was used for detection of GM-CSF production. In brief, cells were stimulated by PMA (50 ng/ml, Sigma) and ionomycin (500 ng/ml, Sigma) for 4 h, except for experiments using compound inhibitors in Fig. 5 and S5, where PMA and ionomycin stimulation were not used. Brefeldin A (2 μg/ml, Sigma) was added for the last 2 h before cells were harvested for analysis. For detection of p-p38, cells were first fixed with the BD Cytofix kit and permeablized with 100% methanol followed by

staining with rabbit–anti-mouse–p-p38 and APC-goat–anti-rabbit-IgG. Flow-cytometry data were collected using the Gallios flow cytometer (Beckman). Sorting of cells were performed on the Moflo Astrios cell sorter.

**Histological analysis**. Tissues from proximal colon were dissected and fixed with 4% paraformaldehyde. Tissues were then embedded in paraffin, sectioned at 5 μm, and stained with H&E. Sections were then blindly analyzed using the light microscope (Olympus), and scored according to a previously described scoring system[66]. The four parameters used include (i) the degree of inflammatory infiltration in the LP, range 1–3; (ii) Goblet cell loss as a marker of mucin depletion, range 0–2; (iii) mucosal erosion to frank ulcerations, range 0–2; and (iv) submucosal spread to transmural involvement, range 0–2.The severity of inflammation in sections of the colon was based on the sum of the scores in each parameter (maximum score = 9).

**Immunofluorescence**. $Rag1^{-/-}Rorc^{gfp/+}$ mice were perfused with 20 ml 1 × PBS followed by 20 ml 4% paraformaldehyde (PFA) in PBS. Large intestines were washed three times with 1 × PBS at 4 °C, cut open, rolled up with inside out, and dehydrated in 30% sucrose overnight at 4 °C. Whole intestinal Swiss rolls were embedded in OCT (Tissue-Tek®) at −80 °C for at least 6 h. The frozen tissue blocks were cut into 5 μm slices. After blocked with 1% BSA, sections were stained with anti-GFP-Alexa Fluor®488 (Thermo Fisher Scientific) and 4,6 diamidino-2-phenylindole (DAPI, Thermo Fisher Scientific). Sections were then observed on a fluorescence microscope Zeiss Axio Imager A2. Areas of cryptopatches were analyzed by ImageJ software (National Institutes of Health, Bethesda, MD, USA). Density of ILC3s in observed cryptopatch was calculated by the area of crypto-patches divided by the number of GFP+ cells.

**RNA-seq analysis**. About $2 \times 10^5$ sorted large intestinal ILC3s (Lin−GFP+ cells) were pooled from 3–6 mice and lysed in Trizol (Invitrogen). The total RNA was extracted. Biological duplicates were generated for each group. The total RNA sample is digested by DNaseI (NEB), and purified by oligo-dT beads (Dynabeads mRNA purification kit, Invitrogen), then poly(A) containing mRNA were fragmented into 130 bp with First-strand buffer. First-strand cDNA is generated by N6 primer, First Strand Master Mix and Super Script II reverse transcription (Invitrogen) (reaction condition: 25 °C for 10 min; 42 °C for 40 min; 70 °C for 15 min). Then add Second Strand Master Mix to synthesize the second-strand cDNA (16 °C for 1 h). Purified the cDNA with Ampure XP Beads (AGENCOURT), then combine with End Repair Mix, incubate at 20 °C for 30 min. Purified and add A-Tailing Mix, incubate at 37 °C for 30 min. Then combine the Adenylate 3′ ends DNA, Adapter and Ligation Mix, incubate the ligate reaction at 20 °C for 20 min. Several rounds of PCR amplification with PCR Primer Cocktail and PCR Master Mix were performed to enrich the cDNA fragments. Then the PCR products were purified with Ampure XP Beads (AGENCOURT). The Qualified libraries will amplify on cBot to generate the cluster on the flowcell (TruSeq PE Cluster Kit V3–cBot–HS, Illumina). The amplified flowcell will be sequenced pair end on the HiSeq 2000 System (TruSeq SBS KIT-HS V3, Illumina), read length 50. Reads were mapped to Mouse Genome Assembly GRCm38.p5 by STAR v2.5. Gene and isoform expression quantification was called by RSEM v1.2 with default parameters on GENCODE mouse M16 gene annotation file. Differential expression analysis was performed by Bioconductor package edgeR v3.18.1. Significantly changed genes were chosen according to two criteria: (1) significance level $p < 0.05$; (2) expression level average FPKM values bigger than 1 in either treatment or control groups. Significantly changed genes for more than 1.5-fold, which were used for gene ontology enrichment analysis using the website of Gene Ontology Consortium (http://www.geneontology.org), were further filtered with the following criteria: (1) significance level $p < 0.05$, FDR < 0.25; (2) expression level average FPKM values bigger than 5 in either treatment or control groups. (3) Fold change of mean expression between a-DR3 and control group is more than 1.5. Heatmap was generated with software HemI 1.0. Normalized heatmap was based on the standard score (Z score) and generated with software HemI 1.0. The standard score of a raw score $x$ is $Z = \frac{x-\mu}{\sigma}$, where $\mu$ is the mean of the FPKM value of each sample and σ is the standard deviation of the FPKM value of each sample.

**Human samples**. Tonsil tissues were cut into 3–10 -mm fragments, and mechanically disrupted using the plunger end of a plastic syringe. Cell suspensions were filtered through a 70-μm cell strainer, and mononuclear cells were isolated with Lymphoprep™ (Axis Shield). Human tonsils were from tonsillectomies, and informed consent was obtained from all participants in this study. The study was approved by the Independent Ethics Committee of Shanghai Tongren Hospital (approval number 2016-020-01).

**Reagents**. α-CD40 (FGK4.5) was from Bioxcel. DSS was purchased from MP biomedicals. Z-VAD-FMK was from Apexbio Technology. Compound inhibitors used for in vitro study were SB 203580 (Selleck Chemicals), PF-3644022 (Tocris), SP600125 (Selleck Chemicals), Bay11-7085 (Selleck Chemicals), and SCH772984 (Apexbio Technology). Recombinant human TL1A was from Peprotech. DR3-Fc was synthesized by Biointron Biological Inc. Briefly, DR3-Fc was constructed by conjugating extracellular domain of mouse DR3 with mouse IgG1. The plasmid was purified from transiently transfected supernatant of the HEK293F cells by Protein A affinity column.

**Quantitative real-time RT-PCR**. RNA was isolated with Trizol reagent (Invitrogen). cDNA was synthesized using the GoScript™ Reverse Transcription kit (Promega). Real-time PCR was performed using SYBR Green (Bio-rad). Reactions were run with the Mx 3000 P Q-PCR System (Angilent). The results were displayed as relative expression values normalized to β-actin. A list of primers used for real-time RT-PCR is presented in Supplementary Table 2.

**Statistical methods**. Unless otherwise noted, statistical analysis was performed with the Mann–Whitney U test on individual biological samples using GraphPad Prism 5.0 program. The data from such experiments are presented as mean values ± SEM; $*p < 0.05$ was considered statistically significant; $**p < 0.01$; $***p < 0.001$. For analyses where Student's $t$ tests were used, Klomogorov–Smirnov test was performed to confirm data are normally distributed.

**Reporting summary**. Further information on research design is available in the Nature Research Reporting Summary linked to this article.

## Data Availability

The source data for Figs. 1B-J, 1L-N, 1P-R, 2A-E, 2G, 2H, 2J, 2K, 2N, 3A-C, 3E-O, 4A-G, 5A-G, 5I, 6A, 6B, 6D, 6E, 6G, 6H-J, 7A-Q and supplementary Figures 1A, 1B, 1G, 1H, 1J, 1K, 2B, 2C, 2E, 2F, 5A-D, 5F-H, 6B-F, 7A-G, 8A, 8B, 8D, 8K are provided as a Source Data File. RNA-seq data have been deposited in the Gene Expression Omnibus database under the accession code GSE114546. Other data that support the findings of this study are available from the corresponding author upon reasonable request.

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

## Acknowledgements

The authors would like to thank Dr. Yang-xin Fu, Dr. Jiawei Zhou, Dr. Lei Shen, Dr. Mingzhao Zhu, Dr. Xi Chen, and Dr. Liming Sun for the experimental support. We would like to thank Dr. Lei Fang, Dr. Zhengyi Wang, and Dr. Changchun Zha for suggestions on the project. We thank the entire J.Q. laboratory for their help and suggestions. This study was supported by grants 2015CB943400 and 2014CB943300 from the Ministry of Science and Technology of China, grant XDB19000000 from the "Strategic priority research program of the Chinese Academy of Sciences", grants 91542102 and 31570887 from the National Natural Science Foundation of China and China's Youth 1000-Talent Program to J.Q. and grant 81672083 from the National Natural Science Foundation of China to H.S.

## Author contributions

J.Q., J.L., and W.S. designed the research. J.L. and W.S. conducted the experiments and analyzed the data. H. Sun, Y.J., Y.C., and J.S. helped with experiments. J.Q., J.L. and W.S. wrote the paper. H. Sheng, X.G., L.Z., and T.C. supported the project and assisted in revising the paper. J.Q. supervised the project.
