## [Peer Review File · Nature Communications]

Reviewers' comments:

Reviewer #1 (Remarks to the Author):

In this paper the authors investigated the effects of TL1A/DR3 signaling on ILC3s. They report that activation of DR3 signaling by an agonistic antibody (α -DR3) enhanced GM-CSF production from intestinal ILC3s through the p38 pathway. GM-CSF subsequently caused accumulation of eosinophils, neutrophils and CD11b⁺CD11c⁺ myeloid cells, which "expulsed" ILC3s from the intestine through an IL-23-dependent manner and exacerbated colitis. Blockade of either GM-CSF or IL-23 sufficiently prevented α -DR3-driven ILC3 loss. However, IL-23 could "expulse" ILC3s in the absence of GM-CSF, suggesting that IL-23 acts downstream of GM-CSF in this process. Neutralization of TL1A by soluble DR3 ameliorated α -CD40-induced colitis while α -DR3 worsened DSS-induced colitis independently of the adaptive immune system. Importantly, no deleterious effects of α -DR3 on DSS-induced innate colitis was observed in the absence of ILC3s, suggesting a pivotal role for ILC3s in mediating the effects of α -DR3. The authors conclude that their findings dissects the process and consequence of DR3 signaling-induced intestinal inflammation through regulation of ILC3s.

This paper contains a significant amount of work using state-of-the-art models and methodologies. It contains significant mechanistic and novel information regarding the the effects of DR3 signaling on ILC3 and intestinal inflammation. However, there are significant problems with this paper:

General comments:

- 1) The paper is poorly written with many grammatical mistakes. In addition, the authors use unconventional terminologies that makes the manuscript difficult to read and comprehend. For example, they frequently refer to the term "expulsion" of ILC3s from the intestine. it is unclear what they mean by that. Their results show that the loss of ILC3s driven by α -DR3 is not due to cell apoptosis or fate conversion but they don't certainly show that ILC3s are expelled from the gut into the intestinal lumen. Similarly, they repeatedly use the term "agonizing" antibody. What does that mean? Perhaps they mean agonistic antibody.
- 2) The Discussion section does not really address many of the results presented or other studies in the literature with different results. For example, a recent paper showed that DR3 deficiency exacerbates the severity of DSS-induced colitis. The effects of the α DR3 antibody in DSS-induced colitis presented in this study should be discussed in the context of this paper. In this context the authors should also perform DR3 blockade in DSS-induced colitis and α DR3 stimulation in CD40 colitis to further prove their hypothesis.
- 3) The authors do not satisfactorily address scientific rigor and data reproducibility, in general, and the role of the microbiome, in particular. In the Experimental procedures section that should report in details how they controlled for housing artifacts, effects of the gut microbiome on DSS colitis, randomization methodologies for the in vivo studies, diet, light cycle, bedding etc.

Specific comments:

Line 1: the verb "expulse" is confusing. I suggest to rephrase the sentence for clarity

Line 42: I believe the author means agonist Ab to DR3 and not "agonizing". Please, change this throughout the manuscript

Line 74: Please add reference doi: 10.3389/fimmu.2018.00362

Line 111: I believe again that the author means agonist Ab to DR3 and not "agonizing"

Line 112: the verb "expulse" is confusing. I suggest to rephrase the sentence for clarity

Line 117: "expulse".. Maybe the author means "reduced/contracted/diminished/eliminated", Please, consider change this throughout the manuscript

Line 182: Is it possible that the time point chosen masks the ILC3-ILC1 conversion?

Figures:

The title of the axis is sometimes capital or small letter, please uniform it throughout the manuscript.

Results:

Line 163-170: This whole paragraph is indicated in the title but the Figs are all in the Supplementary material. it is quite difficult to follow. I suggest to put some of the main finding in the main text.

Line 172-182: Why the author chose 4 days after α -DR3 injection to measure ILCs? What id the kinetic of these effects/

Throughout the result section is not always clear where the cells the author analyzed come from,

please specify if the cells are LPL or MLN etc..

Line 242: Eosinophil and neutrophil markers are not specified.

Line 253: Is not specified where the mRNA was isolated from..LPL?

Line 306/Fig 5G: Missing multiple sample analysis for this graph

Line 319: Images seem to be taken at different zoom and with different image setting

Reviewer #2 (Remarks to the Author):

The manuscript by Li and colleagues describes the role of DR3-mediated regulation of ILC3 in intestinal inflammation.

Major comments:

1. Results, page 7, fig 1 and fig S1. Please show the complete gating strategy and FMOs. Definition of ILC1 (Lin⁻ RORg⁻ NKP46⁺) might be including NK cells. This should be explained.
2. Treatment with anti-DR3 antibody clearly diminishes the absolute number and percentage of total Lin⁻ cells of both NKp46⁺ – ILC3. The authors clearly demonstrate that this reduction is not due to increased apoptosis. The authors also suggest that there is no plastic conversion into ILC1 (fig S2). The latter is not fully investigated and it would be interesting to see the percentages of ILC1, ILC2 and non-ILC lin⁻ in all further experiments (figs 2, 3 and 6, and suppl. S5). Particularly in figure 3 after IL-23 treatment some conversion from ILC3 to ILC1 should be expected. Furthermore, the fact that ILC3 just “vanish” from the gut is not discussed enough. Some speculation of their fate and the significance of the finding should be added.
3. The conclusion that IL-23 acts downstream of GM-CSF is not supported by the data shown (results, page 11 and 12). Other studies suggest that those 2 factors act as in an autocrine positive feedback loop (Pearson, eLife, 2016) as is acknowledged by the authors in the discussion. Please rewrite results in a more balanced manner.
4. Fig 2D: treatment with anti-GM-CSF partially prevents the loss of total ILC3 but values do not reach those obtained in PBS control. Please state or discuss what other mechanisms might be involved in this and add p-value between PBS and anti-GM-CSF.

Minor comments:

1. I would suggest to have the manuscript language edited by a native English speaker.
2. Acronyms should be written in brackets after the full word.
3. Methods page 23, Flow cytometry: please add a table or list of antibodies used (clone, fluorochrom, vendor)

Reviewer #3 (Remarks to the Author):

In this manuscript, Li et al study the consequences of acutely stimulating the TNF-family receptor DR3 with an agonist mAb on colonic innate lymphoid cells and murine models of colitis. They show that DR3 agonism stimulates production of GM-CSF and IL-23, which results in depletion of ILC3 from the colonic tissue. In the anti-CD40 model of colitis, DR3 antagonism through administration of DR3-Fc resulted in relative protection from weight loss and colonic pathology, and in DSS colitis, agonistic DR3 mAb resulted in enhanced pathology and weight loss. Previous work has characterized the loss of ILC3 as being pathogenic in innate immune models of colitis, and whereas blockade or genetic deficiency of TL1A or DR3 in DSS colitis have had mixed effects. Previous work has not connected DR3 signaling to the GM-CSF/IL-23 axis in colitis so in that respect this work is novel

The main weakness of this work in terms of contributing to the understanding of IBD pathogenesis is that the majority of the experiments are performed with acute stimulation with an agonistic DR3 mAb or hydrodynamic injection of TL1A which are non-physiological stimuli. The experiments showing that DR3 blockade ameliorates anti-CD40 mediated colitis are helpful to show a pathogenic role for endogenous TL1A-DR3 interactions in pathogenesis of intestinal inflammation. However, more experiments need to be done to show that the pathway defined by DR3 agonistic antibodies is operable for physiological TL1A-DR3 interactions, such as showing that ILC3 produce GM-CSF in a DR3-dependent manner.

Addition specific issues which need to be addressed are listed below

Major issues

1. The authors make a case that ILC3 depletion in the colon is the result of altered trafficking of these cells rather than conversion to ILC1 or apoptosis. However, data to support migration to another anatomical location is lacking. Other tissues such as Mesenteric LN, small intestine, and spleen should be studied to identify tissues where the ILC3 are migrating to after acute agonism of DR3.
The authors should show an isotype or FMO control rather than Lin⁻ cells as negative control for DR3 expression
2. There are also unresolved conflicts between this work and past studies e.g. Jia et al, J Immunol 2016) who showed DR3 deficient mice having enhanced disease. These need to be resolved in the discussion
3. Figure 3: The authors show data implicating IL-23 in the depletion of ILC3 in the colon after anti-DR3 mAb treatment. However, p40 blockade neutralizes both IL-12 and IL-23. Although hydrodynamic injection of IL-23 rather than IL-12 plasmid reproduced this phenomenon, a blocking experiment with anti-IL-23 specific mAb against the p19 subunit should be performed to demonstrate the necessity for IL-23 in TL1A-mediated effects. The authors should also show the absolute numbers as shown in the other figures, not just the percentage of ILC3.
4. Figure 5. The complete RNAseq dataset used to generate the data shown in Figure 5A needs to be included as supplementary data and deposited into GEO. Cellular viability needs to be shown for the SB p38 inhibitor to confirm that the reduced GM-CSF expression is not due to toxic effects of the inhibitor as well as the absolute numbers, and the effects of anti-DR3 mAb should be confirmed with recombinant TL1A for mouse cells as is done for the human lymphocytes.
5. Figure 6H&I: Control experiments with administration of PBS and DR3Fc in the absence of anti-CD40 should be shown, so that the degree of depletion of these cells by anti-CD40 and any spontaneous effects of DR3 blockade can be seen in comparison to the two conditions shown.
6. Figure 7: absolute numbers of neutrophils, eosinophils and ILC3 should be shown too
7. Statistical methods: The pairwise comparisons should be analyzed for significance with non-parametric (Mann-Whitney) tests rather than the t-test, because of the small numbers of repeats and the non-normally distributed data points where they are shown.

Minor Issues

1. It should be clarified in the text and figure legends whether GFP or RORgt intracellular staining is used to quantitate RORgt-expressing cells in the flow cytometry experiments with Rorc gfp/+ mice
2. Figure 3D-E: The panels should be plotted in the same way, with a preference for the dot-plot shown in 3E rather than the 'plunger' plots shown in 3D
3. The manuscript should be further edited for grammatical correctness.

Responses to Comments

We wish to thank all the referees for their constructive comments on our manuscript, “Activation of DR3 signaling causes loss of ILC3s and exacerbates intestinal inflammation” (“expulses” has been changed to “causes loss of” according to reviewer 1’s suggestion). To address all the referees’ questions, we performed additional experiments to strengthen our manuscript, resulting in 14 additional figure panels in the main figure (Figures 1M, 2H, 3F-3J, 3L, 5F, 5I, 6H, 7A-7C), 14 panels in the supplementary figures (Figures S1C, S1E, S2E, S2F, S5A-S5G, S6A-S6C) and 4 figures for the reviewers (Figures R1-R4). RNA-seq data used for analysis in Figure 5 and S4 have been added as Tables S1. List of antibodies used for flow cytometry has been added as Table S2. Supplementary table legends were added in supplementary information. Axis scale including numbers along the axis was added for all the representative flow cytometry plots.

Specific points from the reviewers mentioned in the comments from the editor has been addressed in answer to “general comments 3)” from reviewer 1 (animal housing and potential other confounding influences), answer to “question 2” from reviewer 2 (possible fate of ILC3s) and answer to “general comments” from reviewer 3 (physiological reflection of TL1A-DR3 interactions) respectively.

Point-to-point responses were provided below to answer every comment from reviewers. The manuscript has been sent for language editing through the “nature publishing group service” for corrections of grammatical mistakes. The changes that we have made in the text of the manuscript are highlighted in yellow, except for some minor corrections for grammatical mistakes or typos. We hope the scope, importance and novelty of the findings will make the revised manuscript appropriate for publication in *Nature Communications*.

Reviewers' comments:

Reviewer #1 (Remarks to the Author):

In this paper the authors investigated the effects of TL1A/DR3 signaling on ILC3s. They report that activation of DR3 signaling by an agonistic antibody (α -DR3) enhanced GM-CSF production from intestinal ILC3s through the p38 pathway. GM-CSF subsequently caused accumulation of eosinophils, neutrophils and CD11b+CD11c+ myeloid cells, which "expulsed" ILC3s from the intestine through an IL-23-dependent manner and exacerbated colitis. Blockade of either GM-CSF or IL-23 sufficiently prevented α -DR3-driven ILC3 loss. However, IL-23 could "expulse" ILC3s in the absence of GM-CSF, suggesting that IL-23 acts downstream of GM-CSF in this process. Neutralization of TL1A by soluble DR3 ameliorated α -CD40-induced colitis while α -DR3 worsened DSS-induced colitis independently of the adaptive immune system. Importantly, no deleterious effects of α -DR3 on DSS-induced

innate colitis was observed in the absence of ILC3s, suggesting a pivotal role for ILC3s in mediating the effects of α -DR3.

The authors conclude that their findings dissect the process and consequence of DR3 signaling-induced intestinal inflammation through regulation of ILC3s.

This paper contains a significant amount of work using state-of-the-art models and methodologies. It contains significant mechanistic and novel information regarding the the effects of DR3 signaling on ILC3 and intestinal inflammation. However, there are significant problems with this paper:

We would like to thank the reviewer for the above comments.

General comments:

1) The paper is poorly written with many grammatical mistakes. In addition, the authors use unconventional terminologies that makes the manuscript difficult to read and comprehend. For example, they frequently refer to the term "expulsion" of ILC3s from the intestine. it is unclear what they mean by that. Their results show that the loss of ILC3s driven by α -DR3 is not due to cell apoptosis or fate conversion but they don't certainly show that ILC3s are expelled from the gut into the intestinal lumen. Similarly, they repeatedly use the term "agonizing" antibody. What does that mean? Perhaps they mean agonistic antibody.

We would like to thank the reviewer for suggestions and corrections. We have refined the writing of our manuscript using the language editing service from nature publishing group. Some major changes in the descriptions, such as “worsening colitis” changed to “exacerbation of colitis”, were highlighted in the manuscript.

We agree that the word “expulsion” was not appropriately used without evidence showing ILC3s migrating out of the lamina propria. Therefore, we have changed the description of “expulsion” throughout the manuscript. For the term “agonizing”, we used it as “agonizing DR3”, which means activation of DR3 signaling. When we refer to the antibody, we do have used the description of “agonistic antibody” rather than “agonizing antibody” in the manuscript. To avoid any misinterpretations, we change the description of “agonizing DR3” to “activation of DR3 signaling” throughout the manuscript.

The changed descriptions are shown below:

ID	position	original description	changed description
1	Line 1	expulses	causes loss of
2	Line 33	expulsed	induced loss of
3	Line 35	expulse	induce loss of
4	Line 37	agonizing DR3	α -DR3
5	Line 43	agonizing DR3	α -DR3
6	Line 43	expulses ILC3s	eliminates ILC3s
6	Line 62	agonizing DR3	Activation of DR3 signaling

7	Line 105	agonizing DR3	α -DR3
8	Line 106	expulsing	induced loss of ILC3s
9	Line 111	expulsed	could eliminate
10	Line 189	expulsion of ILC3s	loss of ILC3s
11	Line 210	expulse ILC3s	induce reduction of ILC3s
12	Line 212	expulsion of ILC3s	loss of ILC3s
13	Line 215	expulsion of ILC3s	elimination of ILC3s
14	Line 234	expulsion of ILC3s	reduction of ILC3s
15	Line 375	agonizing DR3	activation of DR3 signaling
16	Line 380	agonizing DR3	activation of DR3 signaling
17	Line 382	expulsed from	eliminated from
18	Line 416	expulses ILC3s	induces loss of ILC3s
19	Line 426	expulsion	exclusion

2) The Discussion section does not really address many of the results presented or other studies in the literature with different results. For example, a recent paper showed that DR3 deficiency exacerbates the severity of DSS-induced colitis. The effects of the aDR3 antibody in DSS-induced colitis presented in this study should be discussed in the context of this paper. In this context the authors should also perform DR3 blockade in DSS-induced colitis and aDR3 stimulation in CD40 colitis to further prove their hypothesis.

We would like to thank the suggestions from the reviewer. In the introduction, we have mentioned that DR3 or TL1A-deficient mice have been found to be susceptible to DSS-induced colitis (Page 3, Line 59)(Jia et al., 2016). However, another study showed that blockade of TL1A ameliorated chronic DSS-induced colitis by suppressing Th1 and Th17 responses(Takedatsu et al., 2008). To further determine the role of TL1A in innate ILC responses in intestinal inflammation, we blocked TL1A with DR3-Fc in *Rag1*^{-/-} mice in DSS-induced colitis. We found mice treated with DR3-Fc lost less weight and had dampened intestinal inflammation, as indicated by increased colon length and less infiltration of neutrophils during DSS-induced colitis (Figures 7A-7C and S6C). Our data have also shown that activation of DR3 signaling using DR3 agonistic antibody exacerbated DSS-induced colitis in both wild-type and *Rag1*^{-/-} mice (Figures 7D and 7I). Collectively, in both wild-type and *Rag1*^{-/-} mice, TL1A/DR3 signaling plays a pathogenic role in DSS-induced colitis.

The above data seem to be contradictory with the study using TL1A or DR3-deficient mice(Jia et al., 2016). Genetic deficiency of TL1A or DR3 may cause defects in non-immune system, epithelial cell integrity for example, during development. Since this may be a process requiring long time or occurring at an earlier age of the mice, transient blockade of TL1A may not be able to recapitulate the situation in TL1A or DR3 knockout mice. Genetic deficiency in TL1A/DR3 could be considered to be a long-term prevention protocol for colitis, which may not be completely mimicked by transient blockade of TL1A/DR3 (the treatment protocol). It is also possible that antibody treatment leaves marginal amount of TL1A which might be beneficial for

controlling inflammation by supporting Tregs or other mechanisms. It will be necessary to test whether a prevention protocol by long-term blockade of TL1A/DR3 using antibodies would increase the susceptibility to intestinal inflammation.

In the discussion we have added:

Page 19, Line 389:

“A previous study showed that TL1A or DR3-deficient mice were susceptible to DSS-induced colitis due to impaired maintenance of Tregs(Jia et al., 2016). Our data, together with another study, suggests that TL1A/DR3 signaling plays a pathogenic role in DSS-induced colitis(Takedatsu et al., 2008). The discrepancies among the findings may be due to differential consequences mediated by transient blockade of TL1A/DR3 signaling using antibodies, versus complete loss of TL1A/DR3 signaling by genetic ablation. Genetic deficiency of TL1A or DR3 could be considered to be a long-term prevention protocol for colitis, which may affect immune or non-immune system developmentally and couldn't be completely represented with antibody blockade. It is likely that physiological level of TL1A/DR3 is important for homeostasis of intestinal immunity under the steady-state. In the future, it will be necessary to test whether a prevention protocol by long-term blockade of TL1A/DR3 using antibodies would increase the susceptibility to intestinal inflammation. ”

Data for reviewer (not included in the manuscript):

We also tested if α -DR3 exacerbated α -CD40 colitis in *Rag1*^{-/-} mice. Loss of ILC3s occurred as early as 24 hr after α -CD40 treatment even in the absence of α -DR3 injection (Figure R1A). Significant loss of ILC3s would make any exacerbation effect of α -DR3 on α -CD40-induced colitis, by targeting ILC3, less likely to happen. We therefore treated mice with control IgG or α -DR3 only once 1 day before α -CD40 injection (Figure R1B).

A mildly but significantly more weight loss was observed in α -DR3 treated group compared to control group at day 1 upon α -CD40 treatment in *Rag1*^{-/-} mice (Figure R1B). However, control group quickly caught up with α -DR3-treated group on day 2 (Figure R1B). The severity of disease was similar in control and α -DR3-treated group when mice were sacrificed on day 2, as indicated by similar colon lengths and histological scores from the two groups (Figure R1C-R1F). The above data suggest α -DR3 has limited effect on exacerbation of α -CD40-induced colitis.

Compared to DSS-induced colitis where control group didn't have obvious weight loss until day 4, mice of control group developed acute and dramatic weight loss within 2 days in α -CD40-induced colitis (Figure R1B). The acute course of the disease was accompanied by significant upregulation of TL1A expression (Figure 6A). The effect of endogenous TL1A on α -CD40-induced colitis may be saturated and a further effect on exacerbation of α -CD40-induced colitis by α -DR3 is minimal.

Figure R1 α -DR3 treatment has limited effect on exacerbation of α -CD40-induced colitis.

(A) $Rag1^{-/-}$ mice were treated with 50ug α -CD40. Large intestinal lamina propria lymphocytes (LPLs) were isolated 24 hr later. Percentages of ILC3s ($Lin^{-}ROR\gamma^{+}$ cells) gated on Lin^{-} (CD3, B220, CD11b, CD11c negative) cells were analyzed by flow cytometry and shown. (B-F) $Rag1^{-/-}$ mice were treated with isotype IgG or α -DR3 (1ug) 24 hr before mice were injected with 50ug of α -CD40. (B) Percentages of weight change from indicated groups were calculated and shown. (C) Lengths of colons from indicated groups were shown. (D) Representative image of colon from indicated groups were shown. (E) Paraffin-embedded colon sections were stained with hematoxylin and eosin (H&E). Scale bar was 100um. (F) Histological scores were evaluated and shown. (A, B, C and F) Data are means \pm SEM. (A-F) Data are representative of two independent experiments.

3) The authors do not satisfactorily address scientific rigor and data reproducibility, in general, and the role of the microbiome, in particular. In the Experimental procedures section that should report in details how they controlled for housing artifacts, effects of the gut microbiome on DSS colitis, randomization methodologies for the in vivo studies, diet, light cycle, bedding etc.

We thank the reviewer for the suggestion. We have confirmed that we state how many times experiments have been repeated in all the figures.

“In DSS-induced colitis, α -CD40-induced colitis and other in vivo experiments, littermate mice were co-housed since weaning and randomly separated into control and treatment groups to avoid variation in distribution of microflora. All mice were fed with a plain commercial diet (Silaikang, Shanghai). Mice were housed in corn-cob-bedding cages in a room with light-dark cycle (lights on at 6:00 and off at 18:00).” We have added the above description in the “Mice” of “Methods” section (Page 22, Line 458).

Specific comments:

Line 1: the verb “expulse” is confusing. I suggest to rephrase the sentence for clarity

We have rephrased the sentence, please see answer to general points 1).

Line 42: I believe the author means agonist Ab to DR3 and not “agonizing”. Please, change this throughout the manuscript

We have rephrased the sentence, please see answer to general points 1).

Line 74: Please add reference doi: 10.3389/fimmu.2018.00362

We have added the above reference.

Line 111: I believe again that the author means agonist Ab to DR3 and not “agonizing”

We have rephrased the sentence, please see answer to general points 1).

Line 112: the verb “expulse” is confusing. I suggest to rephrase the sentence for clarity

We have rephrased the sentence, please see answer to general points 1).

Line 117: “expulse”.. Maybe the author means “reduced/contracted/diminished/eliminated”, Please, consider change this throughout the manuscript

We have rephrased the sentence, please see answer to general points 1).

Line 182: Is it possible that the time point chosen masks the ILC3-ILC1 conversion?

We examined the kinetics of loss of ILC3s after α -DR3 treatment. Loss of ILC3s was initially found on day 4. For details, please see answer to the question below “Line 172-182: Why the author chose 4 days after a-DR3 injection to measure ILCs? What is the kinetic of these effects”.

Figures:

The title of the axis is sometimes capital or small letter, please uniform it throughout the manuscript.

We have uniformed the titles for the axes as “first letter capitalized and small letters for the rest of them”.

Results:

Line 163-170: This whole paragraph is indicated in the title but the Figs are all in the

Supplementary material. it is quite difficult to follow. I suggest to put some of the main finding in the main text.

We removed previous “Figures S2A, S2B and S2G” to main “Figures 2A-2C”.

Line 172-182: Why the author chose 4 days after α -DR3 injection to measure ILCs? What is the kinetic of these effects/

Data for reviewer:

We injected α -DR3 to *Rag1*^{-/-} mice for a longer period of time every other day for 8 days. We examined the kinetics of numbers of ILC3s on day 1, 2, 3, 4 and 8 upon α -DR3 treatment (Figure R2). We found numbers of ILC3s were even lower on day 8 of α -DR3 treatment than day 4, whereas there was no significant difference in numbers of ILC3s through day 1 to day 3 compared to day 0 (Figure R2A). Therefore, we considered day 4 to be a time point when loss of ILC3s consistently started and continued afterwards.

As suggested by reviewer 2’s question 2, we also analyzed numbers of ILC2s, group 1 ILCs (g1-ILCs, the nomenclature was modified following the suggestion in question 1 from reviewer 2) and Lin⁻non-ILCs. We found no compensatory increase in numbers of g1-ILCs, ILC2s and Lin⁻non-ILCs on day 1-3 post α -DR3 treatment. This data suggest that it is unlikely for ILC3s to convert to the above cell populations in α -DR3-treated mice on day 1-3 (Figure R2B). Therefore, day 4 is a reasonable time point for us to choose to analyze the fate conversion of ILC3s in α -DR3-treated mice.

Figure R2 Kinetics of ILCs during α -DR3 treatment. *Rag1*^{-/-} mice were treated with 1 μ g of α -DR3 every other day. Large intestinal LPLs were isolated. Percentages of ILC3 (Lin⁻ROR γ t⁺), ILC2 (Lin⁻GATA3⁺), group 1 ILCs (Lin⁻ROR γ t⁻NKp46⁺) and Lin⁻non-ILCs (Lin⁻ROR γ t⁻GATA3⁻NKp46⁻) were analyzed by flow cytometry. (A) Numbers of ILC3s at different time points after α -DR3 treatment was calculated and shown. (B) Numbers of ILC2s, g1-ILCs and Lin⁻non-ILCs at different time points after α -DR3 treatment was calculated and shown. (A and B) Data are means \pm SEM. Data are representative of two independent experiments.

Throughout the result section is not always clear where the cells the author analyzed come from, please specify if the cells are LPL or MLN etc..

All the data we got were analysis of large intestinal LPLs. We specified this in the first paragraph of the result section and didn’t repetitively describe it in the following

manuscript. We apologize that we didn't specify it clearly in some of the figure legends in the previous version of the manuscript. For those places, we have confirmed that we have added and highlighted the description that the cells were from large intestinal LPLs.

Line 242: Eosinophil and neutrophil markers are not specified.

Page 12, Line 243 and 245:

We have specified markers for identification of eosinophils (CD11b⁺CD11c⁻Siglec-F⁺) and neutrophils (CD11b⁺CD11c⁻Ly6G⁺).

Line 253: Is not specified where the mRNA was isolated from..LPL?

We have clarified in the figure legends of Figure 4 that the mRNA was extracted from large intestinal epithelial cells (IECs) and different myeloid populations purified from large intestinal LPLs (Page 33, Line 704 and 706).

Line 306/Fig 5G: Missing multiple sample analysis for this graph

We calculated the mean fluorescence intensity (MFI) of phosphorylated p38 (p-p38) in all the groups (Figure 5I). Fold of p-p38 MFI relative to non-stimulated control was calculated (Figure 5I). Statistical analysis revealed significant elevated level of p-p38 in α -DR3-stimulated cells at 30min and 1 hr, as well as in PMA and ionomycin-stimulated cells compared to control group (Figure 5I).

We have also added this data in the result section:

Page 15, Line 305:

“Consist with this observation, an increased level of phosphorylated p38 was observed in ILC3s within 1 hour of α -DR3 treatment (Figures 5H and 5I).”

Line 319: Images seem to be taken at different zoom and with different image setting.

We are sorry for the unsatisfying quality of the figure. We have replaced the representative image with a new one (see new version of Figure 6F).

Reviewer #2 (Remarks to the Author):

The manuscript by Li and colleagues describes the role of DR3-mediated regulation of ILC3 in intestinal inflammation.

Major comments:

1. Results, page 7, fig 1 and fig S1. Please show the complete gating strategy and FMOs.

Definition of ILC1 (Lin⁻ RORg- NKP46⁺) might be including NK cells. This should be explained.

We thank the reviewer for the above suggestions. Gating strategies for Figures 1A-1E and Figures 1G-1I have been provided in Figures S1C and S1E. FMO control for DR3 staining has been shown in Figure S1D.

We apologize for the inaccurate definition for ILC1s. Following a recently published standard nomenclature of ILCs, we have changed the description of “ILC1s” to “group 1 ILCs (g1-ILCs, including ILC1s and NK cells)” (Vivier et al., 2018; Wong et al., 2018). And we have also clarified that the g1-ILCs described in our manuscript doesn’t include B220⁺/CD11b⁺/CD11c⁺ NK cells (Page 7, Line 134).

Label for Figure 1D, S1D has been changed accordingly. And we have explained in our manuscript as following.

Page 7, Line 134:

“As a note, the g1-ILCs (classically composed of NK cells and ILC1s) that we described in our study didn’t include B220⁺/CD11b⁺/CD11c⁺ NK cells (Vivier et al., 2018; Wong et al., 2018).”

2. Treatment with anti-DR3 antibody clearly diminishes the absolute number and percentage of total Lin⁻ cells of both NKp46⁺ – ILC3. The authors clearly demonstrate that this reduction is not due to increased apoptosis. The authors also suggest that there is no plastic conversion into ILC1 (fig S2). The latter is not fully investigated and it would be interesting to see the percentages of ILC1, ILC2 and non-ILC lin⁻ in all further experiments (figs 2, 3 and 6, and suppl. S5). Particularly in figure 3 after IL-23 treatment some conversion from ILC3 to ILC1 should be expected. Furthermore, the fact that ILC3 just “vanish” from the gut is not discussed enough. Some speculation of their fate and the significance of the finding should be added.

We thank the reviewer for bringing up the above points. We thoroughly examined proportions and cell numbers of ILC2, g1-ILCs (see answer to question 1) and non-ILC Lin⁻ cells in α-DR3 treatment, GM-CSF hydrodynamic injection and IL-23 hydrodynamic injection experiments. In above experiments, there was a trend towards a proportionally increased ILC2s but not g1-ILCs or Lin⁻non-ILC cells gated on Lin⁻ cells (Figure S2E, R3A and R3B). However, total numbers of ILC2s/g1-ILCs/Lin⁻ non-ILC cells were not significantly changed in all the above models (Figure S2F, R3C and R3D). These data, together with data of Figures 2C and S2D, collectively indicate that fate conversion of ILC3 to ILC2s/g1-ILCs/non-ILC Lin⁻ cells is less likely to be the reason for loss of ILC3s in α-DR3-treated mice. We therefore didn’t further analyze percentage/numbers of ILC2s/g1-ILCs/non-ILC Lin⁻ cells in α-CD40-induced colitis or DSS-induced colitis.

Figure S2E and S2F have been added. In the text, we have added:

Page 9, Line 175:

“Consistently, no compensatory increase in numbers of g1-ILCs, ILC2s or Lin⁻non-ILCs was found upon α -DR3 treatment, although there was trend towards a proportionally increased level of ILC2s among Lin⁻ cells probably due to primary loss of ILC3s (Figures S2E and S2F).”

Data for reviewer:

Figure R3 Overexpression of GM-CSF or IL-23 has minimal effect on level of non-ILC3-ILCs. *Rag1*^{-/-} mice were injected with 10ug of GM-CSF or IL-23-expressing plasmid or control plasmid by hydrodynamic injection. Large intestinal LPLs were isolated 4 days later. Percentages of ILC2s (Lin⁻GATA3⁺), group 1 ILCs (g1-ILCs, Lin⁻ROR γ t⁻NKp46⁺ cells) and Lin⁻non-ILCs (Lin⁻ROR γ t⁻GATA3⁻NKp46⁻ cells) were analyzed by flow cytometry. Percentages (A and B) and total numbers (C and D) of ILC2s, g1-ILCs and Lin⁻non-ILCs (Lin⁻ROR γ t⁻GATA3⁻NKp46⁻) were shown. (A-D) Data are means \pm SEM. Data are representative of two independent experiments.

Speculations on the fate of ILC3s have been added in the discussion:

Page 21, Line 429:

“We demonstrated that loss of ILC3s induced by α -DR3 is less likely due to apoptosis or fate conversion in situ. It is possible that ILC3s undergo other forms of cell death independently of caspase 3, such as necroptosis (Kearney and Martin, 2017). Mice with genetic deficiency for key signaling molecules controlling other forms of cell death will help to test this possibility. Search for ILC3s in multiple tissues failed to locate accumulation of ILC3s in the small intestine, mesenteric lymph nodes, spleen, peritoneal cavity or blood (data not shown). Although fate conversion of ILC3s was not found in situ, it is also possible that ILC3s firstly migrate to other organs and then fate conversion of ILC3s takes place. A thorough spatial-temporal inspection for ILC3s locally and in various organs, especially with the facilitation of Kaede mouse and intravital microscopy, is required for further elucidation of the mechanisms for loss of ILC3s (Mackley et al., 2015; Tomura et al., 2008).”

Some significance of the finding has been discussed in the first 2 paragraphs of the discussion section on potential risk of clinical application of α -DR3. Based more findings in the revised version of this manuscript, we have also rewritten the paragraph of “The loss of ILC3s driven by α -DR3 is probably a negative feedback mechanism to limit the overt inflammation....” to discuss the significance of loss of ILC3s during colitis as the following:

Page 21, Line 440:

“A previous study suggests that mobilization of ILC3s out of the cryptopatches may play a role in coordination and perpetuation of the inflammation in the gut (Pearson et al., 2016). However, final reduction of ILC3s can result in contraction of the inflammation, which works as an efficient feedback mechanism to control overt inflammatory responses. The strategies we used for blocking loss of ILC3s using α -GM-CSF, α -p19, α -p40 or DR3-Fc in this study may also simultaneously ameliorate colitis by limiting the pathogenicity of ILC3s or other mechanisms (Pearson et al., 2016; Uhlig et al., 2006). This makes it difficult for us to assess the consequence of loss of ILC3s per se, without affecting cytokine production of ILC3s. One would suspect abnormal retention of proinflammatory ILC3s would lead to persistent inflammation and severe tissue damage, which would be important to investigate in the future.”

3. The conclusion that IL-23 acts downstream of GM-CSF is not supported by the data shown (results, page 11 and 12). Other studies suggest that those 2 factors act as in an autocrine positive feedback loop (Pearson, eLife, 2016) as is acknowledged by the authors in the discussion. Please rewrite results in a more balanced manner.

We would like to thank the reviewer’s advice. We agree with the reviewer. Indeed, both IL-23 and GM-CSF was increased in α -DR3-treated mice (Figures 2D and 3C). Therefore, it is more likely IL-23 and GM-CSF act in an autocrine manner to eliminate ILC3s in α -DR3-treated mice. We therefore modified the conclusion as following:

Page 12, Line 237:

“This suggests that overexpression of IL-23 could induce loss of ILC3s in a GM-CSF-independent mechanism.”

Page 13, Line 261:

“A previous study has shown that IL-23 promotes GM-CSF production from ILC3s (Pearson et al., 2016). Since both IL-23 and GM-CSF were upregulated by α -DR3 treatment (Figures 2D and 3C), it is likely that IL-23 and GM-CSF collaboratively drive the loss of ILC3s in an autocrine loop during α -DR3-induced inflammation.”

Descriptions as “IL-23 acts downstream of GM-CSF” in the previous version of the manuscript were deleted in the abstract, introduction and results sections.

Page 11, Line 214:

Subtitle was changed from “IL-23 acts downstream of GM-CSF to induce loss of ILC3s upon α -DR3 treatment” to “*IL-23 is important for α -DR3-induced loss of ILC3s*”.

”

4. Fig 2D: treatment with anti-GM-CSF partially prevents the loss of total ILC3 but values do not reach those obtained in PBS control. Please state or discuss what other mechanisms might be involved in this and add p-value between PBS and anti-GM-CSF.

We have added p-value for statistical analysis of percentages of ILC3s in Lin⁻ cells in α -DR3+ α -GM-CSF group compared to PBS group (Figures 2G and 2H). Statistical analysis was performed with Mann-Whitney test as suggested by reviewer 3 question 7. Indeed, a trend towards a reduction in percentages of ILC3s was observed in α -DR3+ α -GM-CSF group compared to PBS group (p=0.06 was added, Figure 2G). We also calculated absolute numbers of ILC3s in the three groups. We found no significant reduction of ILC3s in α -DR3+ α -GM-CSF group compared to PBS-treated group (p=0.29) (Figure 2H). Furthermore, we have shown that blockade of GM-CSF sufficiently suppressed the expression of IL-23, which is important for α -DR3-induced loss of ILC3s. We therefore conclude that blocking GM-CSF prevents α -DR3-induced loss of ILC3s. And possible GM-CSF-independent mechanisms, though likely exist, play limited role in α -DR3-induced loss of ILC3s.

Minor comments:

1. I would suggest to have the manuscript language edited by a native English speaker. **The manuscript language has been edited by nature publishing group service. Some major corrections in the descriptions, such as “worsening colitis” changed to “exacerbation of colitis”, were highlighted in the manuscript.**

2. Acronyms should be written in brackets after the full word. **We have confirmed that acronyms were placed in brackets after the full word in the manuscript. Changes that have been made were highlighted in the revised manuscript.**

3. Methods page 23, Flow cytometry: please add a table or list of antibodies used (clone, fluorochrom, vendor) **We have provided a list of flow cytometry antibodies used in this study as Table S2.**

Reviewer #3 (Remarks to the Author):

In this manuscript, Li et al study the consequences of acutely stimulating the TNF-family receptor DR3 with an agonist mAb on colonic innate lymphoid cells and murine models of colitis. They show that DR3 agonism stimulates production of GM-CSF and IL-23, which results in depletion of ILC3 from the colonic tissue. In the anti-CD40 model of colitis, DR3 antagonism through administration of DR3-Fc resulted in relative protection from weight loss and colonic pathology, and in DSS colitis, agonistic DR3 mAb resulted in enhanced

pathology and weight loss. Previous work has characterized the loss of ILC3 as being pathogenic in innate immune models of colitis, and whereas blockade or genetic deficiency of TL1A or DR3 in DSS colitis have had mixed effects. Previous work has not connected DR3 signaling to the GM-CSF/IL-23 axis in colitis so in that respect this work is novel

The main weakness of this work in terms of contributing to the understanding of IBD pathogenesis is that the majority of the experiments are performed with acute stimulation with an agonistic DR3 mAb or hydrodynamic injection of TL1A which are non-physiological stimuli. The experiments showing that DR3 blockade ameliorates anti-CD40 mediated colitis are helpful to show a pathogenic role for endogenous TL1A-DR3 interactions in pathogenesis of intestinal inflammation. However, more experiments need to be done to show that the pathway defined by DR3 agonistic antibodies is operable for physiological TL1A-DR3 interactions, such as showing that ILC3 produce GM-CSF in a DR3-dependent manner.

We thank the reviewer for comments on our paper. To further support administration of DR3 agonistic antibody is operable for physiological TL1A-DR3 interaction in colitis, we also performed DR3-Fc treatment in DSS-induced colitis in *Rag1*^{-/-} mice (Figures 7A-7C), in supplement to our data showing exacerbation of the disease by α -DR3 (Figures 7D-7Q). We found DR3-Fc-treated mice had ameliorated disease compared to control group (Figures 7A-7C). For details, please see answer to “general comments 2)” from reviewer 1. The above data suggest that endogenous TL1A is pathogenic in DSS-induced colitis in *Rag1*^{-/-} mice. We therefore deleted the paragraph discussing the pathogenicity of ILC3s in DSS-induced colitis with or without α -DR3 treatment in the discussion section.

We also analyzed GM-CSF expression by ILC3s in α -CD40-induced colitis with or without DR3-Fc treatment (Figure 6H). There were variations in the percentages of GM-CSF expression in ILC3s within both DR3-Fc-treated and IgG-treated group on day 1 post α -CD40 injection. Since littermate mice were used in this experiment, we compared level of GM-CSF expression in ILC3s from each littermate pairs using “Wilcoxon matched pairs test”. A significant reduction was consistently observed in GM-CSF expression by ILC3s from DR3-Fc-treated group compared to controls. This data is consistent with the data showing effect of α -DR3 on promoting GM-CSF production by ILC3s. Taken together, we conclude DR3 agonistic antibody is operable for physiological TL1A-DR3 interaction in colitis.

Figures 6H, 7A-7C, S6C have been added. In the result section, we have added:

Page 16, Line 324:

“Consistent with the function of α -DR3 in promoting GM-CSF expression by ILC3s, we observed reduced GM-CSF expression by ILC3s in DR3-Fc-treated mice 24 hr after α -CD40 injection (Figure 6H).”

Page 16, Line 336:

“A previous study showed that blockade of TL1A ameliorated chronic DSS-induced colitis by suppression of Th1 and Th17 responses in wild-type mice (Takedatsu et al., 2008). In

DSS-induced colitis in Rag1^{-/-} mice, we found that DR3-Fc-treated mice lost less weight and had increased colon length (Figures 7A, 7B and S6C). Total Numbers of neutrophils were less in DR3-Fc treated mice (Figure 7C). This suggests that TL1A/DR3 signaling is pathogenic in DSS-induced colitis in the absence of adaptive immune system. Consistent with the protective effect of DR3-Fc in colitis,...”

Addition specific issues which need to be addressed are listed below

Major issues

1. The authors make a case that ILC3 depletion in the colon is the result of altered trafficking of these cells rather than conversion to ILC1 or apoptosis. However, data to support migration to another anatomical location is lacking. Other tissues such as Mesenteric LN, small intestine, and spleen should be studied to identify tissues where the ILC3 are migrating to after acute agonism of DR3.

The reviewer has raised an important point. Time course analysis revealed no significant reduction of ILC3s until day 4 upon α -DR3 treatment (see answer to “Why the author chose 4 days after α -DR3 injection to measure ILCs?” from reviewer 1). We therefore checked ILC3s 4 days after α -DR3 treatment in multiple organs/tissues including the lamina propria of small intestine, spleen, mesenteric lymph nodes and the peritoneal cavity in *Rag1^{-/-}* mice (Figure R4). Barely any ILC3s were found in the peripheral blood on day 4 after α -DR3 treatment (Figure R4A). Intriguingly, we also didn’t detect accumulation of ILC3s in the above analyzed organs, as indicated by both percentages and numbers of ILC3s (Figure R4B). Rather, decreased ILC3s was also observed in the spleen upon α -DR3 treatment (Figure R4B).

Figure for reviewer:

Figure R4 No accumulation of ILC3s in multiple tissues after α -DR3 treatment. *Rag1^{-/-}* mice were treated with 1ug α -DR3 every other day. Small intestinal LPLs, lymphocytes from peripheral blood (PBMC), mesenteric lymph nodes (MLN), spleen and peritoneal

lavage fluid were isolated on day 4 upon α -DR3 treatment. (A) Expression of lineage markers (CD3, B220, CD11b and CD11c) and ROR γ t gated on live lymphocytes from PBMC was analyzed by flow cytometry. Percentages of ILC3s (Lin⁻ROR γ t⁺) in Lin⁻ cells were shown. (B) Percentages and numbers of ILC3s (Lin⁻ROR γ t⁺) from indicated groups were calculated and shown. Data are means \pm SEM. (A and B) Data are representative of two independent experiments.

Previous study has reported that in α -CD40-induced colitis, increased mobilization of ILC3s occurred and was mediated by GM-CSF (Pearson et al., 2016). In the above study, reduction of ILC3s has been consistently observed. However, whether this is due to enhanced motile activity of ILC3s is unclear (Pearson et al., 2016). α -DR3-induced inflammation shares similarities with α -CD40-induced colitis in terms of increased level of GM-CSF and IL-23, accompanied with loss of ILC3s. Transcriptome sequencing data indicated that ILC3s in α -DR3 treated mice had increased expression of genes belonging to the category of locomotion (Figure 5A and S4A). However, a direct proof of migration of ILC3s following α -DR3 treatment requires a thorough spatial-temporal inspection for ILC3s locally and in various organs, especially with the facilitation of *Kaede* mice and intravital microscopy (Tomura et al., 2008). This is difficult to be achieved within the scope of this manuscript and will be interesting for further exploration in the future.

We have added speculations of the fate of ILC3s in the discussion as below. Page 21, Line 429:

*“We demonstrated that loss of ILC3s induced by α -DR3 is less likely due to apoptosis or fate conversion in situ. It is possible that ILC3s undergo other forms of cell death independently of caspase 3, such as necroptosis (Kearney and Martin, 2017). Mice with genetic deficiency for key signaling molecules controlling other forms of cell death will help to test this possibility. Search for ILC3s in multiple tissues failed to locate accumulation of ILC3s in the small intestine, mesenteric lymph nodes, spleen, peritoneal cavity or blood (data not shown). Although fate conversion of ILC3s was not found in situ, it is also possible that ILC3s firstly migrate to other organs and then fate conversion of ILC3s takes place. A thorough spatial-temporal inspection for ILC3s locally and in various organs, especially with the facilitation of *Kaede* mouse and intravital microscopy, is required for further elucidation of the mechanisms for loss of ILC3s (Mackley et al., 2015; Tomura et al., 2008).”*

The authors should show an isotype or FMO control rather than Lin⁻ cells as negative control for DR3 expression

We have provided an FMO control for DR3 staining and modified Figure S1D.

2. There are also unresolved conflicts between this work and past studies e.g. Jia et al, J Immunol 2016) who showed DR3 deficient mice having enhanced disease. These need to be resolved in the discussion.

We would like to thank the reviewer for this suggestion. We discussed the mentioned study in the discussion. For details, please see answer to “general comments 2) The Discussion section does not really address many of the results presented or other studies in the literature with different results.” from reviewer 1.

3. Figure 3: The authors show data implicating IL-23 in the depletion of ILC3 in the colon after anti-DR3 mAb treatment. However, p40 blockade neutralizes both IL-12 and IL-23. Although hydrodynamic injection of IL-23 rather than IL-12 plasmid reproduced this phenomenon, a blocking experiment with anti-IL-23 specific mAb against the p19 subunit should be performed to demonstrate the necessity for IL-23 in TL1A-mediated effects.

We thank the reviewer for the above suggestions. To elucidate if IL-12 or IL-23 is essential for loss of ILC3s induced by α -DR3, we blocked p19 (subunit of IL-23) or p75 (IL-12 heterodimer) with neutralizing antibodies in α -DR3-treated mice (Figures 3G-3J). Loss of ILC3s could be partially rescued by neutralization of p19 but not p75, as indicated by percentages of ILC3s in Lin⁻ cells (Figures 3G-3J). The effect on preventing α -DR3-induced loss of ILC3s was less for α -p19 than α -p40, probably due to differential efficiency of antibodies. Absolute numbers of ILC3s were similar with or without α -p19 treatment probably due to variations in numbers of ILC3s from different batches of experiments (data not shown). Average absolute numbers of ILC3s per mouse harvested from each batch of experiment was significantly more in α -p19 but not α -p75-treated group compared to control group (Figures 3I and 3J). The above data suggest IL-23 is important for α -DR3-induced ILC3 loss.

In the results section we have added (Page 11, Line 224):

“Consistently, blockade of IL-23 with α -p19, but not blockade of IL-12 with α -p75 neutralizing antibodies, reversed loss of ILC3s induced by α -DR3, as indicated by percentages and absolute numbers of ILC3s (Figures 3G-3J).”

Although α -p19 significantly reversed α -DR3-induced loss of ILC3s, the efficiency was less than α -p40. This may be due to differential blocking efficacies of the two neutralizing antibodies. It could also be possible that IL-12 and IL-23 play redundant roles in α -DR3-induced loss of ILC3s. If this is the case, IL-12-driven loss of ILC3s occurs only when mice were treated with α -DR3 but not under the steady-state, which was supported by the fact that overexpression of IL-12 doesn't eliminate ILC3s from the large intestine (Figures 3K and 3L). Based on this hypothesis, it is likely that some IL-12-responsive cells, which could induce the loss of ILC3s, were recruited to the large intestine by α -DR3 treatment. Alternatively, some cells that don't respond to IL-12 under the steady-state were induced to express the receptor for IL-12 by α -DR3 and subsequently induced loss of ILC3s. Cells expressing receptor for IL-12, including ILC3s, ILCs, NK cells, myeloid cells and epithelial cells, could be involved in this mechanism (Klose et al., 2013; O'Sullivan et al., 2016; Regoli et al., 2018). And IL-12 may lead to loss of ILC3s through a cell-intrinsic or extrinsic manner upon α -DR3 treatment.

A thorough elaboration of signaling cascades underlying this possible mechanism requires the facilitation of cellular, molecular and genetic tools, and is beyond the scope of the current study.

Nevertheless, our data explicitly support that IL-23 is critical in α -DR3-induced loss of ILC3s. Firstly, blockade of p19 reversed loss of ILC3s induced by α -DR3. Secondly, blockade of IL-12 by α -p75 had no effect on α -DR3-induced loss of ILC3s. Thirdly, overexpression of IL-23 sufficiently eliminated ILC3s from the large intestine. Since the effect of α -p19 on preventing loss of ILC3s was not as substantial as blockade of p40, we modified the following descriptions in abstract and the result section.

Abstract:

Page 2, Line 34:

“Blockade of either GM-CSF or IL-23 sufficiently prevented α -DR3-driven ILC3 loss” was changed to “*Blockade of either GM-CSF or IL-23 reversed α -DR3-driven ILC3 loss*”.

Introduction:

Page 5, Line 110”:

“Blockade of either GM-CSF or IL-23 could prevent α -DR3 driven ILC3 loss, ...” was changed to “*Blockade of either GM-CSF or IL-23 could reverse α -DR3-driven ILC3 loss,...*”

The authors should also show the absolute numbers as shown in the other figures, not just the percentage of ILC3.

Figures 3F and 3L showing absolute numbers of ILC3s in p40 neutralization experiment and IL-23 overexpression experiment have been added.

4. Figure 5. The complete RNAseq dataset used to generate the data shown in Figure 5A needs to be included as supplementary data and deposited into GEO.

The RNA-seq data that we have used for analysis for Figure 5A have been included in the supplementary data (Table S1) in the manuscript. List of significantly changed genes with FPKM value has been deposited into GSE114546. We noticed a mistake in the number of genes, the expression of which was significantly changed. We are sorry for this mistake and we have changed it: Page 13, Line 269:

“516 and 610 genes were found to be upregulated and downregulated” was changed to “*515 and 609 genes were found to be upregulated and downregulated*”.

We have also corrected and described in detail in the methods section for the criteria of choosing differentially expressed genes and protocol for gene ontology analysis:

Page 27, Line 568:

“*Significantly changed genes were chosen according to two criteria: 1) significance level $p < 0.05$; 2) expression level average FPKM values bigger than 1 in either treatment or control groups. Significantly changed genes for more than 1.5 fold, which were used for*

gene ontology enrichment analysis using the website of Gene Ontology Consortium (<http://www.geneontology.org>), were further filtered with the following criteria: 1) significance level $p < 0.05$, $FDR < 0.25$; 2) expression level average FPKM values bigger than 5 in either treatment or control groups. 3) Fold change of mean expression between α -DR3 and control group is more than 1.5.”

Cellular viability needs to be shown for the SB p38 inhibitor to confirm that the reduced GM-CSF expression is not due to toxic effects of the inhibitor as well as the absolute numbers, and the effects of anti-DR3 mAb should be confirmed with recombinant TL1A for mouse cells as is done for the human lymphocytes.

SB203580 has no cytotoxic effect to human ILC3s or mouse ILC3s, as indicated by no reduction of percentages of live cells in lymphocytes, as well as percentages of ILC3s gated on live lymphocytes in SB203580-treated group compared to other control groups when cells were harvested for analysis (Figure S5A-S5F). Similar to α -DR3, TL1A enhanced GM-CSF production by ILC3s, which could be suppressed by SB203580 (Figure 5F). And no cytotoxicity of TL1A or SB203580 on ILC3s was observed, as indicated by similar absolute numbers of ILC3s when cells were harvested for analysis in all groups (Figure S5G).

In the result section, we have added (Page 15, Line 301):

“And TL1A-induced GM-CSF expression by ILC3s was similarly inhibited by SB203580 (Figure 5F).”

Page 15, Line 306:

“SB203580 had no cytotoxic effects on human or mouse ILC3s, as indicated by the comparable percentages of live lymphocytes, similar number of ILC3s and no reduction of proportions of ILC3s among lymphocytes in SB203580-treated groups compared to control groups (Figure S5). Therefore, the suppression of GM-CSF expression in ILC3s by SB203580 was not due to cytotoxicity.”

Figures 5F and S5 have been added accordingly.

5. Figure 6H&I: Control experiments with administration of PBS and DR3Fc in the absence of anti-CD40 should be shown, so that the degree of depletion of these cells by anti-CD40 and any spontaneous effects of DR3 blockade can be seen in comparison to the two conditions shown.

We agree with the reviewer. We treated mice with or without DR3-Fc at 200ug per day for 3 days. The percentages and total numbers of ILC3s in the large intestinal LPLs were analyzed by flow cytometry (Figures S6A and S6B). No difference in percentages and numbers of ILC3s was found between IgG versus DR3-Fc treated group under the steady-state. Therefore, the enhanced number of ILC3s in DR3-Fc-treated group in α -CD40 colitis is not due to possible spontaneous effects of DR3 blockade on number of ILC3s. The above data was included in the manuscript as Figure S6A and S6B.

In the result section, we have added (Page 16, Line 330): “..., whereas no effect of DR3-Fc on percentages and numbers of ILC3s was observed under the steady-state (Figures S6A and S6B).”

6. Figure 7: absolute numbers of neutrophils, eosinophils and ILC3 should be shown too

We agree that it is necessary to show absolute numbers of above cell populations. To make the figure less crowded, we replaced previous Figures 7C, 7E, 7J, 7M showing percentages of cells with new Figures 7F, 7H, 7M and 7P showing absolute numbers of indicated cells. Figure legends have been modified accordingly.

7. Statistical methods: The pairwise comparisons should be analyzed for significance with non-parametric (Mann-Whitney) tests rather than the t-test, because of the small numbers of repeats and the non-normally distributed data points where they are shown.

We thank the reviewer for the advice. We have changed major method for statistical analysis to Mann Whitney U-test unless noted. Numbers of stars, representing p values in figures, have been changed in all the figures accordingly. More samples in each group have been added for Figures 1K-1M, 2D, 3E, 3M, 4C, 4D, 5D, 5E, 5G, S4C. For analyses where Student’s t-tests were used, we performed Klomogorov-Smirnov test to confirm that the data are normally distributed. For some figures in which the sample sizes are small, we applied paired t-test for statistical analyses. A line connected between each pair, indicating littermate mice from the same batch of experiments was shown in some figures (Figures 2C, 3I, 3J and 6H). No changes of the conclusions in our manuscript were found due to change of the statistical methods.

**We have modified the “Statistical analyses” description in the methods as following:
Page 29, Line 619:**

“Unless otherwise noted, statistical analysis was performed with Mann Whitney U-test on individual biological samples using GraphPad Prism 5.0 program. Data from such experiments are presented as mean values \pm SEM; * $p < 0.05$ was considered statistically significant; ** $p < 0.01$; *** $p < 0.001$. For analyses where Student’s t-tests were used, Klomogorov-Smirnov test was performed to confirm data are normally distributed.”

Minor Issues

1. It should be clarified in the text and figure legends whether GFP or RORgt intracellular staining is used to quantitate RORgt-expressing cells in the flow cytometry experiments with Rorc *gfp/+* mice

For all the flow cytometry data that were analyzed using Rorc^{*gfp/+*} or Rag1^{*-/-*} Rorc^{*gfp/+*} mice in the previous version of the manuscript, we used RORgt intracellular staining

rather than ROR γ t-GFP to identify ILC3s unless noted. Since the sample sizes for Figures 1G-1I were small in the previous version of the manuscript, we replaced the data with new panels using *Rag1*^{-/-} mice rather than *Rag1*^{-/-}*Rorc*^{gfp/+} mice (Figures 1G-1I). Gating strategy was provided as Figure S1E. Descriptions in the text of the result section have been changed accordingly. This data equally support our conclusion that “ α -DR3-induced loss of ILC3s is independent of the adaptive immune system”.

Page 7, Line 140:

*“To determine if α -DR3-driven loss of ILC3s was dependent on adaptive immune system, we administered α -DR3 to *Rag1*^{-/-} mice which lack T and B cells.”*

Page 8, Line 146:

*“Using *Rorc*^{gfp/+} *Rag1*^{-/-} mice with a GFP reporter to indicate the expression of ROR γ (Eberl et al., 2004), we analyzed the distribution of ILC3s (GFP⁺ cells) in the large intestine by immunofluorescence staining of GFP. In α -DR3-treated mice, we observed a reduction in the number of cryptopatches where ILC3s are typically localized (Figures 1N and 1O).”*

In Figure S2A, we utilized ROR γ t-GFP rather than ROR γ t staining to identify ILC3s. We apologize that we didn't specify this in the previous version of the manuscript. We have noted this on the label of the axis as “ROR γ t-GFP” (Figure S2A), as well as in the figure legends of Figure S2A.

2. Figure 3D-E: The panels should be plotted in the same way, with a preference for the dot-plot shown in 3E rather than the ‘plunger’ plots shown in 3D

We think the reviewer means Figure 3M for the current version of manuscript (Figure 3G for previously version of the manuscript). We have changed Figure 3M to dot plots instead of plunger plots with more samples added to each group.

3. The manuscript should be further edited for grammatical correctness.

The manuscript has been sent for language editing using nature publishing group service. Some major changes in the descriptions, such as “worsening colitis” changed to “exacerbation of colitis”, were highlighted in the manuscript.

References:

- Eberl, G., Marmon, S., Sunshine, M.J., Rennert, P.D., Choi, Y., and Littman, D.R. (2004). An essential function for the nuclear receptor ROR γ (t) in the generation of fetal lymphoid tissue inducer cells. *Nature immunology* 5, 64-73.
- Jia, L.G., Bamias, G., Arseneau, K.O., Burkly, L.C., Wang, E.C., Gruszka, D., Pizarro, T.T., and Cominelli, F. (2016). A Novel Role for TL1A/DR3 in Protection against Intestinal Injury and Infection. *Journal of immunology* 197, 377-386.

Kearney, C.J., and Martin, S.J. (2017). An Inflammatory Perspective on Necroptosis. *Molecular cell* 65, 965-973.

Klose, C.S., Kiss, E.A., Schwierzeck, V., Ebert, K., Hoyler, T., d'Hargues, Y., Goppert, N., Croxford, A.L., Waisman, A., Tanriver, Y., and Diefenbach, A. (2013). A T-bet gradient controls the fate and function of CCR6-ROR γ mat⁺ innate lymphoid cells. *Nature* 494, 261-265.

Mackley, E.C., Houston, S., Marriott, C.L., Halford, E.E., Lucas, B., Cerovic, V., Filbey, K.J., Maizels, R.M., Hepworth, M.R., Sonnenberg, G.F., *et al.* (2015). CCR7-dependent trafficking of ROR γ mat⁺ ILCs creates a unique microenvironment within mucosal draining lymph nodes. *Nature communications* 6, 5862.

O'Sullivan, T.E., Rapp, M., Fan, X., Weizman, O.E., Bhardwaj, P., Adams, N.M., Walzer, T., Dannenberg, A.J., and Sun, J.C. (2016). Adipose-Resident Group 1 Innate Lymphoid Cells Promote Obesity-Associated Insulin Resistance. *Immunity* 45, 428-441.

Pearson, C., Thornton, E.E., McKenzie, B., Schaupp, A.L., Huskens, N., Griseri, T., West, N., Tung, S., Seddon, B.P., Uhlig, H.H., and Powrie, F. (2016). ILC3 GM-CSF production and mobilisation orchestrate acute intestinal inflammation. *eLife* 5, e10066.

Regoli, M., Man, A., Gicheva, N., Dumont, A., Ivory, K., Pacini, A., Morucci, G., Branca, J.J.V., Lucattelli, M., Santosuosso, U., *et al.* (2018). Morphological and Functional Characterization of IL-12R β 2 Chain on Intestinal Epithelial Cells: Implications for Local and Systemic Immunoregulation. *Frontiers in immunology* 9, 1177.

Takedatsu, H., Michelsen, K.S., Wei, B., Landers, C.J., Thomas, L.S., Dhall, D., Braun, J., and Targan, S.R. (2008). TL1A (TNFSF15) regulates the development of chronic colitis by modulating both T-helper 1 and T-helper 17 activation. *Gastroenterology* 135, 552-567.

Tomura, M., Yoshida, N., Tanaka, J., Karasawa, S., Miwa, Y., Miyawaki, A., and Kanagawa, O. (2008). Monitoring cellular movement in vivo with photoconvertible fluorescence protein "Kaede" transgenic mice. *Proceedings of the National Academy of Sciences of the United States of America* 105, 10871-10876.

Uhlig, H.H., McKenzie, B.S., Hue, S., Thompson, C., Joyce-Shaikh, B., Stepankova, R., Robinson, N., Buonocore, S., Tlaskalova-Hogenova, H., Cua, D.J., and Powrie, F. (2006). Differential activity of IL-12 and IL-23 in mucosal and systemic innate immune pathology. *Immunity* 25, 309-318.

Vivier, E., Artis, D., Colonna, M., Diefenbach, A., Di Santo, J.P., Eberl, G., Koyasu, S., Locksley, R.M., McKenzie, A.N.J., Mebius, R.E., *et al.* (2018). Innate Lymphoid Cells: 10 Years On. *Cell* 174, 1054-1066.

Wong, E., Xu, R.H., Rubio, D., Lev, A., Stotesbury, C., Fang, M., and Sigal, L.J. (2018). Migratory Dendritic Cells, Group 1 Innate Lymphoid Cells, and Inflammatory Monocytes Collaborate to Recruit NK Cells to the Virus-Infected Lymph Node. *Cell reports* 24, 142-154.

Reviewers' comments:

Reviewer #1 (Remarks to the Author):

The authors have revised the manuscript following the suggestions of the 3 reviewers and performed a significant amount of additional work. The paper remains an important contribution to the field of TL1A/DR3 and intestinal inflammation and contains significant mechanistic data. Since the initial submission of this paper a manuscript dealing with the role of TL1A/DR3 in experimental IBD has been published. The authors should reference this paper in the Introduction section for completeness.

Butto' et al. *Inflamm Bowel Dis*. 2018 Oct 5. doi: 10.1093/ibd/izy305. [Epub ahead of print]

Reviewer #3 (Remarks to the Author):

In their revision, Li et al effectively address many of the points raised in the initial review. There are some outstanding questions on the revised figure 6H and 7, and in the discussion the authors need to address the recently published results of Castellanos et al. in *Immunity* (<https://doi.org/10.1016/j.immuni.2018.10.014>) which dissect the effects of TL1A on T cells vs ILC3 with tissue specific deletion of TL1A and DR3.

Assessment of the responses to the main points are below

More experiments need to be done to show that the pathway defined by DR3 agonistic antibodies is operable for physiological TL1A-DR3 interactions, such as showing that ILC3 produce GM-CSF in a DR3-dependent manner. **RESPONSE** The authors have addressed this with measurement of GM-CSF in DR3-Fc treated mice in the CD40 induced colitis model (Figure 6H). I appreciated the trend towards reduced GM-CSF production in the DR3-Fc treated mice. However the use of Wilcoxon matched-pair statistics to compare responses of littermates is not appropriate, as there is no basis for choosing pairing of one particular littermate over another. Unpaired analysis e.g. mann-whitney should be done, even if non-significant, and the trend shown reported as such.

The additional experiment with DR3-Fc treatment of RAG1-/- mice induced to develop DSS colitis in Figure 7A-C is also appreciated. However, the figure panel labeling in the legend and text needs to be checked as the legend lettering does not appear to be correct. Also, the protection of DR3-Fc treated mice is only apparent after 7 days, which is a longer time point than day 4 when the other DSS colitis experiments were terminated at day 4. In addition, the control IgG treated RAG-1 deficient mice were beginning to lose weight by day 4, whereas the PBS treated mice in Figure 7D did not lose any weight at day 4. The authors need to discuss these discrepancies.

Major issues

1. The authors make a case that ILC3 depletion in the colon is the result of altered trafficking of these cells rather than conversion to ILC1 or apoptosis. However, data to support migration to another anatomical location is lacking. Other tissues such as Mesenteric LN, small intestine, and spleen should be studied to identify tissues where the ILC3 are migrating to after acute agonism of DR3.

The authors should show an isotype or FMO control rather than Lin- cells as negative control for DR3 expression

RESPONSE This has been adequately addressed in the revision

2. There are also unresolved conflicts between this work and past studies e.g. Jia et al, *J Immunol* 2016) who showed DR3 deficient mice having enhanced disease. **RESPONSE** This has been adequately addressed in the discussion

3. Figure 3: The authors show data implicating IL-23 in the depletion of ILC3 in the colon after anti-DR3 mAb treatment. However, p40 blockade neutralizes both IL-12 and IL-23. Although

hydrodynamic injection of IL-23 rather than IL-12 plasmid reproduced this phenomenon, a blocking experiment with anti-IL-23 specific mAb against the p19 subunit should be performed to demonstrate the necessity for IL-23 in TL1A-mediated effects. The authors should also show the absolute numbers as shown in the other figures, not just the percentage of ILC3. RESPONSE This has been adequately addressed with the new experiments described

4. Figure 5. The complete RNAseq dataset used to generate the data shown in Figure 5A needs to be included as supplementary data and deposited into GEO. Cellular viability needs to be shown for the SB p38 inhibitor to confirm that the reduced GM-CSF expression is not due to toxic effects of the inhibitor as well as the absolute numbers, and the effects of anti-DR3 mAb should be confirmed with recombinant TL1A for mouse cells as is done for the human lymphocytes.

RESPONSE This has been adequately addressed

5. Figure 6H&I: Control experiments with administration of PBS and DR3Fc in the absence of anti-CD40 should be shown, so that the degree of depletion of these cells by anti-CD40 and any spontaneous effects of DR3 blockade can be seen in comparison to the two conditions shown.

RESPONSE This has been adequately addressed

6. Figure 7: absolute numbers of neutrophils, eosinophils and ILC3 should be shown too

RESPONSE This has been adequately addressed

7. Statistical methods: The pairwise comparisons should be analyzed for significance with non-parametric (Mann-Whitney) tests rather than the t-test, because of the small numbers of repeats and the non-normally distributed data points where they are shown. RESPONSE This has been adequately addressed except for the statistical analysis of data in figure 6H as mentioned above

Minor Issues

1. It should be clarified in the text and figure legends whether GFP or RORgt intracellular staining is used to quantitate RORgt-expressing cells in the flow cytometry experiments with *Rorc* gfp/+ mice RESPONSE This has been adequately addressed

2. Figure 3D-E: The panels should be plotted in the same way, with a preference for the dot-plot shown in 3E rather than the 'plunger' plots shown in 3D RESPONSE This has been adequately addressed

3. The manuscript should be further edited for grammatical correctness. RESPONSE This has been adequately addressed

Reviewer #4 (Remarks to the Author):

Li et al report a loss of ILC3 from the large intestine following engagement of the DR3 receptor, which is dependent upon both GM-CSF and downstream release of IL-23 by myeloid cells. This axis contributes to pathogenesis in two models of colitis in mouse, and provides insights which could have potential relevance for our understanding of human inflammatory bowel disease. The authors have addressed many of the concerns raised by the Reviewer's however several issues remain which should be clarified, and which are important for accurate interpretation of the data.

Major points:-

- While loss of ILC3 is shown to be dependent upon both GM-CSF and IL-23 through a series of chimera and blocking studies, precisely how ILC3 are lost remains unclear. The authors demonstrate loss of large intestinal ILC3 is not due to altered proliferation, cell death, conversion to ILC1 or migration to other tissues (Reviewer Figure 4) - thus the reasons for the observed phenotype are not resolved. Moreover, loss of ILC3 is in most cases driven by agonistic antibody or overexpression of inflammatory cytokines, both of which could be considered superphysiological and thus, may not reflect normal responses to these cytokines at homeostasis or during

inflammation. This is an important consideration which should be discussed as it may also explain discrepancies with contradictory findings regarding the effects of TL1A in ILC3 and colitis reported in previously published studies.

- It is notable that a-DR3 treatment does not lead to a comparable phenotype in the small intestine, where ILC3 are more abundant and where conversion to ILC1 has previously been demonstrated (Reviewer figure 4). I would recommend Reviewer Figure 4 is included as supplemental data as frequency and number of ILC3 at these other tissue sites is of direct relevance to the interpretation of the data.

- The authors state a significantly decreased proportion of NKp46-ILC3 "including NCR- ILC3s and lymphoid tissue inducer cells" (Page 7, Line 130), however the data for ILC3 with a LTi-like phenotype is not actually shown in either Figure 1A/1B or Fig S1C as cited in the text. To support this interpretation the authors would have to include LTi-like ILC3 markers such as CCR6, CD4, MHCII etc and also demonstrate loss of this subset specifically. This is important for the interpretation of the data in particular as the authors report a gene signature following a-DR3 that would support an alternative hypothesis whereby NCR+ and NCR- (non-LTi) ILC3 are lost and LTi-like ILC3 are enriched, thus resulting in the enhanced detection of canonical LTi-like ILC3 genes (Figure S4; increased *Lta*, *Ltb*, *Il22*, *Tnfs4* (OX40L), *Ccr6*, *Cxcr5*, *H2-Ab1*). Indeed this would be in line with a recent report that TL1A induces both IL-22 and *Tnfs4/Ox40l* on ILC3 (Castellanos et al, *Immunity* 2019) and the authors own data that MHCII+ ILC3 were enriched (Figure S4E/F).

- Loss of IL-22 is only shown by PCR, ideally this should be confirmed via intracellular cytokine staining on different ILC3 subsets (see points above).

- Additionally, one aspect of the study lacking is whether ILC3-derived pro-inflammatory cytokines (IFN γ , IL-17A, TNF α , IL-22) directly contribute to colitis in the anti-CD40/DSS models and whether they are perturbed by interfering with DR3-signalling and intracellular staining of ILC3 for these cytokines to address this point would improve the manuscript.

Minor points:-

- The nomenclature g1-ILCs is unnecessary. Please use ILC1, which is the consensus terminology.

- The terminology self and nonself is also not appropriate (page 9) - ILC3-intrinsic/extrinsic or autocrine/paracrine etc would be more appropriate terms.

Responses to Comments

We wish to thank all the referees for their constructive comments on our manuscript, “Activation of DR3 signaling causes loss of ILC3s and exacerbates intestinal inflammation”. To address all the referees’ questions, we performed additional experiments to strengthen our manuscript, resulting in 1 additional panel in the main figure (Figures 1J), 8 figure panels in the supplementary figures (Supplementary Figures 4E, 4F and 5A-5F) and 3 figures for the reviewers (Figures R1-R3 in the letter of point-to-point responses). Additional reagents used for flow cytometry has been added to Supplementary Table 2. Point-to-point responses were provided below *in bold* to answer every comment from reviewers. We hope the scope, importance and novelty of the findings will make the revised manuscript appropriate for publication in *Nature Communications*.

Reviewers' comments:

Reviewer #1 (Remarks to the Author):

The authors have revised the manuscript following the suggestions of the 3 reviewers and performed a significant amount of additional work. The paper remains an important contributions to the field of TL1A/DR3 and intestinal inflammation and contains significant mechanistic data. Since the initial submission of this paper a manuscript dealing with the role of TL1A/DR3 in experimental IBD has been published. The authors should reference this paper in the Introduction section for completeness.

Butto' et al. Inflamm Bowel Dis. 2018 Oct 5. doi: 10.1093/ibd/izy305. [Epub ahead of print]

We would like to thank the suggestion from Reviewer #1. The above reference has been added in the introduction section (Line69-70):

“However, TL1A/DR3 signaling has been predominantly proved by numerous studies to be detrimental in IBD^{1,2,3}.

Reviewer #3 (Remarks to the Author):

In their revision, Li et al effectively address many of the points raised in the initial review. There are some outstanding questions on the revised figure 6H and 7, and in the discussion the authors need to address the recently published results of Castellanos et al. in Immunity (<https://doi.org/10.1016/j.immuni.2018.10.014>) which dissect the effects of TL1A on T cells vs ILC3 with tissue specific deletion of TL1A and DR3.

We would like to thank the comments from Reviewer #3. We have addressed the findings from Castellanos et al. in Immunity

(<https://doi.org/10.1016/j.immuni.2018.10.014>) and modified the paragraph discussing different findings on the role of DR3 signaling in DSS-induced colitis in the discussion.

Line 408-425:

“A previous study showed that TL1A or DR3-deficient mice were susceptible to DSS-induced colitis due to impaired maintenance of Tregs⁴. In addition, a recent study reported that mice with genetic deletion of DR3 specifically on ILC3s were more susceptible to DSS-induced colitis, mainly due to decreased IL-22 from ILC3s⁵. Our data, together with a previous report, suggest that TL1A/DR3 signaling plays a pathogenic role in DSS-induced colitis⁶. The discrepancies among the above findings on the role of DR3 in innate colitis may be due to differential experimental settings using genetic deficient mice compared to transient blockade of DR3 signaling using TL1A neutralization antibody or DR3-Fc. Genetic deficiency of TL1A or DR3 could be considered as a long-term prevention protocol for colitis, which couldn't be completely represented with antibody blockade. It is possible that in mice with genetic deficiency of DR3 on ILC3s, constantly suboptimal level of IL-22 from birth makes the individual more susceptible to exacerbated colitis led by lack of IL-22. Whereas in antibody treatment experiments, blockade of TL1A or DR3 started after onset of DSS-induced colitis, before which no possibly impaired epithelial integrity due to lack of IL-22 occurred. Therefore, blockade of TL1A/DR3 using antibodies resulted in ameliorated colitis likely by suppressing the pathogenicity of ILC3s. In the future, it will be necessary to test whether a prevention protocol by long-term blockade of TL1A/DR3 using antibodies would increase the susceptibility to intestinal inflammation.”

Assessment of the responses to the main points are below

More experiments need to be done to show that the pathway defined by DR3 agonistic antibodies is operable for physiological TL1A-DR3 interactions, such as showing that ILC3 produce GM-CSF in a DR3-dependent manner. RESPONSE The authors have addressed this with measurement of GM-CSF in DR3-Fc treated mice in the CD40 induced colitis model (Figure 6H). I appreciated the trend towards reduced GM-CSF production in the DR3-Fc treated mice. However the use of Wilcoxon matched-pair statistics to compare responses of littermates is not appropriate, as there is no basis for choosing pairing of one particular littermate over another. Unpaired analysis e.g. mann-whitney should be done, even if non-significant, and the trend shown reported as such.

We would like to thank the comments from the reviewer. We repeated this experiment, pooled more samples for this dataset and the data passed the normality test. We used unpaired Student's t-test for statistical analysis and found GM-CSF production from ILC3s in mice treated with DR3-Fc was lower than control group. The difference was close to reach a statistical significance (P=0.06). An updated version of Figure 6H was added.

The additional experiment with DR3-Fc treatment of RAG1^{-/-} mice induced to develop DSS colitis in Figure 7A-C is also appreciated. However, the figure panel labeling in the legend and

text needs to be checked as the legend lettering does not appear to be correct. Also, the protection of DR3-Fc treated mice is only apparent after 7 days, which is a longer time point than day 4 when the other DSS colitis experiments were terminated at day 4. In addition, the control IgG treated RAG-1 deficient mice were beginning to lose weight by day 4, whereas the PBS treated mice in Figure 7D did not lose any weight at day 4. The authors need to discuss these discrepancies.

We thank the reviewer for the above comments. We checked the lettering in the figure panel and in the legends and confirmed that they are correct.

Different Lot numbers of DSS was used for experiments performed in Figures 7A-7C versus Figures 7D-7Q, which could be accountable for variation of disease severity. In addition, severity of DSS-induced colitis could also be affected by microbiota from different mouse colonies. However, since we used gender matched and littermate mice in all the experiments in Figure 7, our conclusion would not be affected by the above confounding factors. We also performed experiments and confirmed that the severity of disease was comparable between PBS and control IgG group, suggesting the effect of α -DR3 agonistic antibody or DR3-Fc on DSS-induced colitis was not caused by any non-specific effect of IgG (Figures R1A-R1D).

Figure R1 Control IgG has no effect on DSS-induced colitis.

Littermate *Rag1*^{-/-} mice were fed with DSS in drinking water as indicated in A and C. (A and B) 200ug of mouse IgG or PBS were i.p. administered daily from day 3-6. (C and D) 1ug of hamster IgG or PBS was i.p. injected to mice on day -2, 0 and 2 post DSS treatment. (A and C) Percentages of weight change were shown. (B and D) Lengths of colons were shown.

In Figures 7A-7C, we aimed to test the treatment efficacy of DR3-Fc on DSS-induced colitis, whereas in Figures 7D-7Q, α -DR3-treatment resulted in a fast and acute course of DSS-induced colitis compared to control group. Experiments in Figures 7A-7C and Figures 7D-7Q were designed to test the loss-of-function and gain-of-function of TL1A/DR3 signaling in colitis respectively. To determine cellular mechanisms *ex vivo*,

we terminated the experiments at different time points when the difference between control and treatment groups became solid, clear and stable.

Reviewer #4 (Remarks to the Author):

Li et al report a loss of ILC3 from the large intestine following engagement of the DR3 receptor, which is dependent upon both GM-CSF and downstream release of IL-23 by myeloid cells. This axis contributes to pathogenesis in two models of colitis in mouse, and provides insights which could have potential relevance for our understanding of human inflammatory bowel disease. The authors have addressed many of the concerns raised by the Reviewer's however several issues remain which should be clarified, and which are important for accurate interpretation of the data.

We thank the reviewer for the encouraging comments.

Major points:

1. While loss of ILC3 is shown to be dependent upon both GM-CSF and IL-23 through a series of chimera and blocking studies, precisely how ILC3 are lost remains unclear. The authors demonstrate loss of large intestinal ILC3 is not due to altered proliferation, cell death, conversion to ILC1 or migration to other tissues (Reviewer Figure 4) - thus the reasons for the observed phenotype are not resolved.

We would like to thank the reviewer's suggestions. We have discussed the potential mechanisms for loss of ILC3s in the discussion section (Line454-463): *“We demonstrated that loss of ILC3s induced by α -DR3 is less likely due to apoptosis or fate conversion in situ. It is possible that ILC3s undergo other forms of cell death independently of caspase 3, such as necroptosis⁷. Mice with genetic deficiency for key signaling molecules controlling other forms of cell death will help to test this possibility. Search for ILC3s in multiple tissues failed to locate accumulation of ILC3s in the small intestine, mesenteric lymph nodes, spleen, peritoneal cavity or blood. Although fate conversion of ILC3s was not found in situ, it is also possible that ILC3s firstly migrate to other organs and then fate conversion of ILC3s takes place. A thorough spatial-temporal inspection for ILC3s locally and in various organs, especially with the facilitation of Kaede mouse and intravital microscopy, is required for further elucidation of the mechanisms for loss of ILC3s^{8,9}”*

Moreover, loss of ILC3 is in most cases driven by agonistic antibody or overexpression of inflammatory cytokines, both of which could be considered superphysiological and thus, may not reflect normal responses to these cytokines at homeostasis or during inflammation. This is an important consideration which should be discussed as it may also explain discrepancies with contradictory findings regarding the effects of TL1A in ILC3 and colitis reported in previously published studies.

We understand the reviewer's concern. Indeed, it is likely that α -DR3 agonistic antibody may not completely recapitulate normal responses to TL1A at homeostasis or during inflammation. However, α -DR3 has been proposed as a therapeutic strategy for treatment of GVHD or cancer. In this respect, our research is valuable for understanding the risk for the application of α -DR3.

The above points have been addressed in the discussion.

Line 392-395:

“The activation of DR3 signaling by an agonistic antibody has been proved to be effective in alleviating GVHD and lung allergy in mouse models by promoting Tregs^{10, 11, 12, 13}. And the activation of DR3 signaling has also been indicated to be applied in immuno-cancer therapy by facilitating the CD8⁺T cell responses¹⁴”.

In addition, we showed that blockade of TL1A using DR3-Fc ameliorated innate colitis, prevented loss of ILC3s and reduced GM-CSF production from ILC3s. The above data suggest non-superphysiological TL1A-DR3 signaling plays an important role in causing loss of ILC3s and intestinal inflammation.

The discrepancies among different findings on the role of TL1A/DR3 signaling in DSS-induced colitis have been addressed in the discussion as the following.

Line 408-425:

“A previous study showed that TL1A or DR3-deficient mice were susceptible to DSS-induced colitis due to impaired maintenance of Tregs⁴. In addition, a recent study reported that mice with genetic deletion of DR3 specifically on ILC3s were more susceptible to DSS-induced colitis, mainly due to decreased IL-22 from ILC3s⁵. Our data, together with a previous report, suggest that TL1A/DR3 signaling plays a pathogenic role in DSS-induced colitis⁶. The discrepancies among the above findings on the role of DR3 in innate colitis may be due to differential experimental settings using genetic deficient mice compared to transient blockade of DR3 signaling using TL1A neutralization antibody or DR3-Fc. Genetic deficiency of TL1A or DR3 could be considered as a long-term prevention protocol for colitis, which couldn't be completely represented with antibody blockade. It is possible that in mice with genetic deficiency of DR3 on ILC3s, constantly suboptimal level of IL-22 from birth makes the individual more susceptible to exacerbated colitis led by lack of IL-22. Whereas in antibody treatment experiments, blockade of TL1A or DR3 started after onset of DSS-induced colitis, before which no possibly impaired epithelial integrity due to lack of IL-22 occurred. Therefore, blockade of TL1A/DR3 using antibodies resulted in ameliorated colitis likely by suppressing the pathogenicity of ILC3s. In the future, it will be necessary to test whether a prevention protocol by long-term blockade of TL1A/DR3 using antibodies would increase the susceptibility to intestinal inflammation”.

2. It is notable that α -DR3 treatment does not lead to a comparable phenotype in the small intestine, where ILC3 are more abundant and where conversion to ILC1 has previously been demonstrated (Reviewer figure 4). I would recommend Reviewer Figure 4 is included as supplemental data as frequency and number of ILC3 at these other tissue sites is of direct

relevance to the interpretation of the data.

We would like to thank the reviewer's suggestion. We have included the "previous Reviewer Figure 4" in the manuscript as Supplementary Figures 4E and 4F.

In the Results section, we added (Line 290-293):

"We observed no accumulation of ILC3s in other organs including the small intestine, mesenteric lymph nodes, spleen, peritoneal cavity or blood, suggesting ILC3s didn't preferentially or specifically translocate to one of the above places (Supplementary Figures 4E and 4F)".

3. The authors state a significantly decreased proportion of NKp46-ILC3 "including NCR-ILC3s and lymphoid tissue inducer cells" (Page 7, Line 130), however the data for ILC3 with a LTi-like phenotype is not actually shown in either Figure 1A/1B or Fig S1C as cited in the text. To support this interpretation the authors would have to include LTi-like ILC3 markers such as CCR6, CD4, MHCII etc and also demonstrate loss of this subset specifically. This is important for the interpretation of the data in particular as the authors report a gene signature following α -DR3 that would support an alternative hypothesis whereby NCR⁺ and NCR⁻ (non-LTi) ILC3 are lost and LTi-like ILC3 are enriched, thus resulting in the enhanced detection of canonical LTi-like ILC3 genes (Figure S4; increased *Lta*, *Ltb*, *Il22*, *Tnfsf4* (OX40L), *Ccr6*, *Cxcr5*, *H2-Ab1*). Indeed this would be in line with a recent report that TL1A induces both IL-22 and *Tnfsf4/Ox40l* on ILC3 (Castellanos et al, *Immunity* 2019) and the authors own data that MHCII⁺ ILC3 were enriched (Figure S4E/F).

We would like to thank the reviewer for raising this important point. Classification and nomenclature for subsets of ILC3s have been proposed by several groups. In 2013, subsets of ILC3s have been proposed by International Union of Immunological Societies (IUIS) to be divided into NCR⁺ILC3s, NCR⁻ILC3s and Lti cells¹⁵. In 2019, the same group has redefined the subsets as NKp46⁺ILC3s and NKp46⁻ILC3s, whereas Lti cells were separated from the ILC3 population¹⁶. As pointed by the reviewer, ILC3s could also be grouped to NCR⁺ILC3s, CCR6⁺NKp46⁻ ILC3s (Lti-like ILC3s) and CCR6⁻ NKp46⁻ ILC3s¹⁷. As a note, Lti-like ILC3s are different from Lti cells but phenotypically mirror the fetal Lti cells¹⁷. And the CCR6⁻NKp46⁻ ILC3s have the potential to be converted to the NCR⁺ILC3s^{17,18}. Our description on the NKp46⁻ ILC3s (including NCR⁻ ILC3s and lymphoid tissue inducer cells) followed the classification of ILC3s proposed in 2013¹⁵. We originally meant to clarify that NKp46⁻ ILC3s were composed of NCR⁻ ILC3s and lymphoid tissue inducer cells. We are sorry that the description led to a misunderstanding that both the NCR⁻ILC3s and lymphoid tissue inducer cells were decreased upon α -DR3 treatment.

In order to keep a uniform nomenclature of ILC3s and diligently addressed the reviewer's point, we deleted the description of "(including NCR⁻ ILC3s and lymphoid tissue inducer cells)". We analyzed percentages of NCR⁺ILC3s, CCR6⁺NKp46⁻ ILC3s (Lti-like ILC3s) and CCR6⁻NKp46⁻ ILC3s upon α -DR3 treatment. Indeed, we observed a trend towards a proportional increase of CCR6⁺NKp46⁻ ILC3s and a trend towards a

proportional decrease of NCR⁺ILC3s and CCR6⁻NKp46⁻ ILC3s among total ILC3s. As the reviewer mentioned, this would cause an upregulated expression of Lti-like genes. Nevertheless, we found percentages of OX40L, MHCII and IL-22 was also significantly upregulated within NCR⁺ILC3s. In addition, NCR⁺ILC3s signature genes such as *Cxcr6* were not decreased in α -DR3-treated ILC3s¹⁹. And Lti-like ILC3s signature gene *Gpr183* was not increased in α -DR3-treated ILC3s²⁰. Moreover, as a NCR⁺ILC3 signature gene, *Gzmc* was upregulated in α -DR3-treated ILC3s²¹. Therefore, gene profile of α -DR3-treated ILC3s was not purely a representative of Lti-like ILC3s over NCR⁺ILC3s gene signature, but also a reflection of genes regulated by DR3 signaling *per se*. In the future, it would be important to dissect the role of TL1A/DR3 signaling on different subsets of ILC3s.

Figure R2 Expression of *Cxcr6*, *Gpr183* and *Gzmc* in α -DR3-treated ILC3s analyzed by RNAseq.

Rag1^{-/-}*Rorc*^{gfp/+} mice were treated with 1ug α -DR3 once and large intestinal LPLs were isolated 3 days later. Duplicates mRNA of FACS purified ILC3s (Lin⁻GFP⁺) cells was extracted and subjected to RNA-seq analysis. Representative FPKM values of genes were shown with Z-normalized heatmap.

In the manuscript we have added (Line 300-308), “In accordance with upregulation of *Ccr6* mRNA expression in ILC3s upon α -DR3 treatment, we observed a trend towards a proportional increase of CCR6⁺NKp46⁻ ILC3s and a trend toward a decrease of NCR⁺ILC3s and CCR6⁻NKp46⁻ ILC3s among total ILC3s (Supplementary Figures 5A and 5B). But the absolute numbers of all subsets of ILC3s were reduced (Supplementary Figure 5C). Notably, the upregulation of MHCII expression was mainly on NCR⁺ILC3s and CCR6⁻NKp46⁻ ILC3s (Supplementary Figure 5D), whereas the upregulation of IL-22 expression was mainly on NCR⁺ILC3s and CCR6⁺NKp46⁻ ILC3s (Supplementary Figure 5E). Enhancement of OX40L expression was found on all subsets of ILC3s upon treatment of α -DR3 (Supplementary Figure 5F).”

4. Loss of IL-22 is only shown by PCR, ideally this should be confirmed via intracellular cytokine staining on different ILC3 subsets (see points above).

Flow cytometry staining confirmed that total numbers of IL-22-producing ILC3s were reduced upon α -DR3 treatment.

We would like to thank the reviewer for the above comments. As was addressed in “question 3 from reviewer 4”, increased percentages of IL-22 were found both in NCR⁺ILC3s and in CCR6⁺NKp46⁻ ILC3s. However, numbers of all subsets of ILC3s were reduced upon α -DR3 treatment. Consequently, absolute number of IL-22-producing cells was decreased in α -DR3-treated mice, which was consistent with a reduction of IL-22 mRNA expression among intestinal LPLs.

In the result section, we added (Line 142-143): “Consistently, absolute number of IL-22-producing ILC3s was decreased in α -DR3-treated mice (Figure 1J)”.

5. Additionally, one aspect of the study lacking is whether ILC3-derived pro-inflammatory cytokines (IFN γ , IL-17A, TNF α , IL-22) directly contribute to colitis in the anti-CD40/DSS models and whether they are perturbed by interfering with DR3-signalling and intracellular staining of ILC3 for these cytokines to address this point would improve the manuscript.

We would like to thank the reviewer for the above suggestion. We analyzed the expression of in ILC3s in DR3-Fc treated mice under α -CD40-induced colitis. We found no difference in percentages of IL-22/IL-17/TNF- α /IFN- γ in ILC3s from DR3-Fc-treated mice compared to IgG-treated mice. The above data suggest that the amelioration of innate colitis by DR3-Fc is less likely to be mediated by regulation of IL-22/IL-17/TNF- α /IFN- γ production from ILC3s.

Figure R3 Expression of inflammatory cytokines by ILC3s was not affected by DR3-Fc treatment in α -CD40-induced colitis

Female littermate *Rag1*^{-/-} mice were injected with 50ug of α -CD40. 200ug of DR3-Fc or control IgG were i.p. administered daily from day 0. Mice were sacrificed for analysis 2 days later after α -CD40 injection and large intestinal LPLs were isolated. For detection IL-22, large intestinal LPLs were treated with brefeldin A 2h before cells were harvested for analysis by flow cytometry. Percentages of IL-22⁺ cells in ILC3s (Lin⁻ROR γ ⁺) were shown. For detection IL-17, TNF- α and IFN- γ , large intestinal LPLs were treated with PMA(50ng/ml)/ionomycin(500ng/ml) and brefeldin A(2ug/ml) 4h before cells were harvested for analysis by flow cytometry. Percentages of IL-17⁺ cells, TNF- α ⁺ cells and IFN- γ ⁺ cells in ILC3s were shown. Data are means \pm SEM. Data are representative of at least 2 independent experiments.

Previous study has shown that percentage of IL-22 in ILC3s was decreased in DR3-deficient mice. However, we didn't observe reduction of IL-22 expression in ILC3s in DR3-Fc-treated mice in α -CD40-induced colitis. The discrepancies among the above data may be due to differential strategies for perturbation of TL1A/DR3 signaling using transient blocking antibody compared to genetic deletion. It is likely that the potency of IL-22 production by ILC3s was intrinsically strong, which couldn't be inhibited by transient blockade of TL1A/DR3 signaling. The above points have been addressed in the discussion section. For details, please see answer to question 1 from reviewer 4. Alternatively, it is possible that we failed to catch the time window when there was a

reduction in IL-22 expression in ILC3s after DR3-Fc treatment (As a note, we also didn't observe reduced level of IL-22 in ILC3s at 24h after induction of α -CD40-colitis in DR3-Fc-treated mice, data not shown). A detailed time-course examination of above cytokines in ILC3s of DR3-Fc-treated mice during innate colitis is beyond the scope of the current manuscript. We therefore didn't further analyze the production of above cytokines in ILC3s in DR3-Fc treated mice under DSS-induced colitis either. Moreover, TNF- α and IFN- γ could also be originated from other sources such as macrophages in the intestine. The elaboration of the pathogenicity of ILC3-derived TNF- α /IFN- γ in innate colitis would require specifically deletion of *Tnf/Ifng* in ILC3s, which would be interesting to be investigated in the future.

Minor points:

1. The nomenclature g1-ILCs is unnecessary. Please use ILC1, which is the consensus terminology.

We thank the reviewer for raising this point. We used ILC1 as the nomenclature in the first version of the manuscript. However, it was pointed out by Reviewer 2 and later acknowledged by us that a proportion of NK cells might also be included in the Lin⁻ROR γ ^tNKp46⁺ population.

Question from Reviewer 2:

“Definition of ILC1 (Lin⁻ ROR γ - NKP46⁺) might be including NK cells. This should be explained.”

The answer to the previous question 1 from reviewer 2 was as the following “We apologize for the inaccurate definition for ILC1s. Following a recently published standard nomenclature of ILCs, we have changed the description of “ILC1s” to “group 1 ILCs (g1-ILCs, including ILC1s and NK cells)”^{16, 22}. And we have also clarified that the g1-ILCs described in our manuscript doesn't include B220⁺/CD11b⁺/CD11c⁺ NK cells.”

Line 130-132:

“As a note, the g1-ILCs (classically composed of NK cells and ILC1s) that we described in our study didn't include B220⁺/CD11b⁺/CD11c⁺ NK cells”^{16, 22}

2. The terminology self and nonself is also not appropriate (page 9) ILC3-intrinsic/extrinsic or autocrine/paracrine etc would be more appropriate terms.

We appreciate the reviewer's correction. We have changed the description to “autocrine/paracrine” in both the result and discussion sections (Line 182, 198 and 688).

** See Nature Research's author and referees' website at www.nature.com/authors for information about policies, services and author benefits

This email has been sent through the Springer Nature Tracking System NY-610A-NPG&MTS

Confidentiality Statement:

This e-mail is confidential and subject to copyright. Any unauthorised use or disclosure of its contents is prohibited. If you have received this email in error please notify our Manuscript Tracking System Helpdesk team at <http://platformsupport.nature.com> .

Details of the confidentiality and pre-publicity policy may be found here <http://www.nature.com/authors/policies/confidentiality.html>

Privacy Policy | Update Profile

DISCLAIMER: This e-mail is confidential and should not be used by anyone who is not the original intended recipient. If you have received this e-mail in error please inform the sender and delete it from your mailbox or any other storage mechanism. Springer Nature Limited does not accept liability for any statements made which are clearly the sender's own and not expressly made on behalf of Springer Nature Ltd or one of their agents.

Please note that Springer Nature Limited and their agents and affiliates do not accept any responsibility for viruses or malware that may be contained in this

References:

1. Bamias G, Jia LG, Cominelli F. The tumor necrosis factor-like cytokine 1A/death receptor 3 cytokine system in intestinal inflammation. *Curr Opin Gastroenterol* **29**, 597-602 (2013).
2. Li Z, *et al.* Death Receptor 3 Signaling Controls the Balance between Regulatory and Effector Lymphocytes in SAMP1/YitFc Mice with Crohn's Disease-Like Ileitis. *Front Immunol* **9**, 362 (2018).
3. Butto LF, *et al.* Death-Domain-Receptor 3 Deletion Normalizes Inflammatory Gene Expression and Prevents Ileitis in Experimental Crohn's Disease. *Inflammatory bowel diseases* **25**, 14-26 (2019).
4. Jia LG, *et al.* A Novel Role for TL1A/DR3 in Protection against Intestinal Injury and Infection. *J Immunol* **197**, 377-386 (2016).
5. Castellanos JG, *et al.* Microbiota-Induced TNF-like Ligand 1A Drives Group 3 Innate Lymphoid Cell-Mediated Barrier Protection and Intestinal T Cell Activation during Colitis. *Immunity* **49**, 1077-1089 e1075 (2018).

6. Takedatsu H, *et al.* TL1A (TNFSF15) regulates the development of chronic colitis by modulating both T-helper 1 and T-helper 17 activation. *Gastroenterology* **135**, 552-567 (2008).
7. Kearney CJ, Martin SJ. An Inflammatory Perspective on Necroptosis. *Mol Cell* **65**, 965-973 (2017).
8. Tomura M, *et al.* Monitoring cellular movement in vivo with photoconvertible fluorescence protein "Kaede" transgenic mice. *Proc Natl Acad Sci U S A* **105**, 10871-10876 (2008).
9. Mackley EC, *et al.* CCR7-dependent trafficking of RORgamma(+) ILCs creates a unique microenvironment within mucosal draining lymph nodes. *Nat Commun* **6**, 5862 (2015).
10. Nishikii H, *et al.* DR3 signaling modulates the function of Foxp3+ regulatory T cells and the severity of acute graft-versus-host disease. *Blood* **128**, 2846-2858 (2016).
11. Kim BS, *et al.* Treatment with agonistic DR3 antibody results in expansion of donor Tregs and reduced graft-versus-host disease. *Blood* **126**, 546-557 (2015).
12. Schreiber TH, *et al.* Therapeutic Treg expansion in mice by TNFRSF25 prevents allergic lung inflammation. *J Clin Invest* **120**, 3629-3640 (2010).
13. Madireddi S, *et al.* Regulatory T Cell-Mediated Suppression of Inflammation Induced by DR3 Signaling Is Dependent on Galectin-9. *J Immunol* **199**, 2721-2728 (2017).
14. Slebioda TJ, *et al.* Triggering of TNFRSF25 promotes CD8(+) T-cell responses and anti-tumor immunity. *Eur J Immunol* **41**, 2606-2611 (2011).
15. Spits H, *et al.* Innate lymphoid cells--a proposal for uniform nomenclature. *Nat Rev Immunol* **13**, 145-149 (2013).
16. Vivier E, *et al.* Innate Lymphoid Cells: 10 Years On. *Cell* **174**, 1054-1066 (2018).
17. Melo-Gonzalez F, Hepworth MR. Functional and phenotypic heterogeneity of group 3 innate lymphoid cells. *Immunology* **150**, 265-275 (2017).

18. Klose CS, *et al.* A T-bet gradient controls the fate and function of CCR6-RORgammat+ innate lymphoid cells. *Nature* **494**, 261-265 (2013).
19. Satoh-Takayama N, *et al.* The chemokine receptor CXCR6 controls the functional topography of interleukin-22 producing intestinal innate lymphoid cells. *Immunity* **41**, 776-788 (2014).
20. Emgard J, *et al.* Oxysterol Sensing through the Receptor GPR183 Promotes the Lymphoid-Tissue-Inducing Function of Innate Lymphoid Cells and Colonic Inflammation. *Immunity* **48**, 120-132 e128 (2018).
21. Gury-BenAri M, *et al.* The Spectrum and Regulatory Landscape of Intestinal Innate Lymphoid Cells Are Shaped by the Microbiome. *Cell* **166**, 1231-1246 e1213 (2016).
22. Wong E, *et al.* Migratory Dendritic Cells, Group 1 Innate Lymphoid Cells, and Inflammatory Monocytes Collaborate to Recruit NK Cells to the Virus-Infected Lymph Node. *Cell Rep* **24**, 142-154 (2018).

REVIEWERS' COMMENTS:

Reviewer #4 (Remarks to the Author):

The study from Li et al has been extensively revised and the updated manuscript provides a comprehensive set of experiments that demonstrate loss of ILC3 upon TL1A signalling, dependent upon GM-CSF and IL-23 which proves effective in suppressing inflammation in experimental colitis.

While the authors have not been able to address the reasons as to why IL-23 induces loss of ILC3 in this context, the manuscript is nonetheless of clear interest and relevance for the field and the mechanisms upstream of this loss have been extensively characterized.

As a minor point I would ask the authors again to please consider use of "ILC1" in place of "g1-ILC". For the sake of clarity it is important that authors follow established nomenclature and do not invent new terms. It is appropriate for the authors to define which cells they are referring to - as in the current revised text i.e. 11b+ cNK cells are excluded, but in this study ILC1 refer to RORgt-NKp46+ cells which could still include NK populations in addition to ILC1 and ex-ILC3, however introducing non-standard terminology should be avoided.

Responses to Comments

REVIEWERS' COMMENTS:

Reviewer #4 (Remarks to the Author):

The study from Li et al has been extensively revised and the updated manuscript provides a comprehensive set of experiments that demonstrate loss of ILC3 upon TL1A signalling, dependent upon GM-CSF and IL-23 which proves effective in suppressing inflammation in experimental colitis.

While the authors have not been able to address the reasons as to why IL-23 induces loss of ILC3 in this context, the manuscript is nonetheless of clear interest and relevance for the field and the mechanisms upstream of this loss have been extensively characterized.

As a minor point I would ask the authors again to please consider use of "ILC1" in place of "g1-ILC". For the sake of clarity it is important that authors follow established nomenclature and do not invent new terms. It is appropriate for the authors to define which cells they are referring to - as in the current revised text i.e. 11b+ cNK cells are excluded, but in this study ILC1 refer to RORgt- NKp46+ cells which could still include NK populations in addition to ILC1 and ex-ILC3, however introducing non-standard terminology should be avoided.

We would like to thank reviewer 4 for the above suggestions. We have changed the nomenclature of “g1-ILC” to “ILC1” in the manuscript, labels in figures and the figure legends. Legends and labels for Figure 1D, Supplementary Figures 1D, 2E and 2F have been changed. Changes that had been made in the manuscript have been marked by “track change”, except for the Supplementary Information, which was required to be finalized without highlights for the final submission.